# Neural Fourier Transform: A General Approach to Equivariant Representation Learning

**Masanori Koyama**[1] **Kenji Fukumizu**[2,1] **Kohei Hayashi**[1] **Takeru Miyato**[3,1]
[1]Preferred Networks, Inc. [2]The Institute of Statistical Mathematics [3]University of Tübingen

## Abstract

Symmetry learning has proven to be an effective approach for extracting the hidden structure of data, with the concept of equivariance relation playing the central role. However, most of the current studies are built on architectural theory and corresponding assumptions on the form of data. We propose Neural Fourier Transform (NFT), a general framework of learning the latent linear action of the group without assuming explicit knowledge of how the group acts on data. We present the theoretical foundations of NFT and show that the existence of a linear equivariant feature, which has been assumed ubiquitously in equivariance learning, is equivalent to the existence of a group invariant kernel on the dataspace. We also provide experimental results to demonstrate the application of NFT in typical scenarios with varying levels of knowledge about the acting group.

## 1 Introduction

Various types of data admit symmetric structure explicitly or implicitly, and such symmetry is often formalized with *action* of a *group*. As a typical example, an RGB image can be regarded as a function defined on the set of 2D coordinates $\mathbb{R}^2 \to \mathbb{R}^3$, and this image admits the standard shift and rotation on the coordinates. Data of 3D object/scene accepts SO(3) action (Chen et al., 2021; Yu et al., 2020), and molecular data accepts permutations (Raghunathan and Priyakumar, 2021) as well. To leverage the symmetrical structure for various tasks, equivariant features are used in many applications in hope that such features extract essential information of data.

Fourier transform (FT) is one of the most classic tools in science that utilizes an equivariance relation to investigate the symmetry in data. Originally, FT was developed to study the symmetry induced by the shift action $a \circ f(\cdot) := f(\cdot - a)$ on a square-integrable function $f \in L_2(\mathbb{R})$. FT maps $f$ invertibly to another function $\Phi(f) = \hat{f} \in L_2(\mathbb{R})$. It is well known that FT satisfies $(a \circ \hat{f})(\omega) = e^{-ia\omega}\hat{f}(\omega)$ and hence the equivariant relation $\Phi(a \circ f) = e^{ia\omega}\Phi(f)$. By this equivariant mapping, FT achieves the decomposition of $L_2(\mathbb{R})$ into shift-equivariant subspaces (also called *frequencies/irreducible representations*). This idea has been extended to the actions of general groups, and extensively studied as *harmonic analysis on groups* (Rudin, 1991). In recent studies of deep learning, group convolution (GC) is a popular approach to equivariance (Cohen and Welling, 2017; Finzi et al., 2021; Cohen and Welling, 2014; 2016; Weiler and Cesa, 2019), and the theory of FT also provides its mathematical foundation (Kondor and Trivedi, 2018; Cohen et al., 2018; 2019).

One practical limitation of FT and GC is that they can be applied only when we know how the group acts on the data. Moreover, FT and GC also assume that the group acts linearly on the constituent unit of input (e.g. pixel). In many cases, however, the group action on the dataset may not be linear or explicit. For example, when the observation process involves an unknown nonlinear deformation such as fisheye transformation, the effect of the action on the observation is also intractable and nonlinear (Fig. 1, left). Another such instance may occur for the 2D pictures of a rotating 3D object rendered with some camera pose (Fig. 1, right). In both examples, any two consecutive frames are implicitly generated as $(x, g \circ x)$, where $g$ is a group action of *shift/rotation*. They are clearly not linear transformations in the 2D pixel space. To learn the hidden equivariance relation describing the symmetry of data in wider situations, we must extend the equivariance learning and Fourier analysis to the cases in which the group action on each data may be nonlinear or implicit.

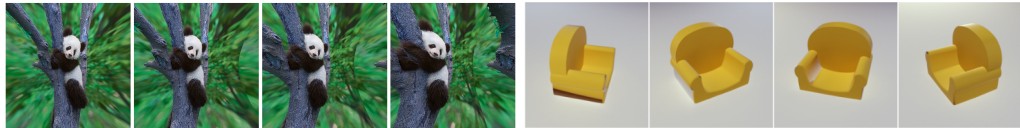

Figure 1: Left: An image sequence produced by applying fisheye transformation after horizontal shifting. Right: 2D renderings of a spinning chair.

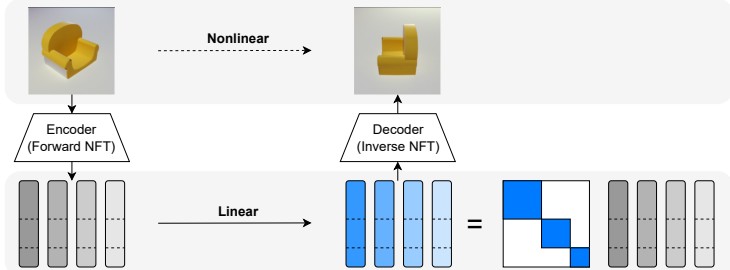

Figure 2: NFT framework. Each block corresponds to irreducible representation/frequency.

To formally establish a solution to this problem, we propose Neural Fourier Transform (NFT), a nonlinear extension of FT, as a general framework for equivariant representation learning. We generalize the approach explored in (Miyato et al., 2022) and provide a novel theoretical foundation for the nonlinear learning of equivariance. As an extension of FT, the goal of NFT is to express the data space as the direct sum of linear equivariant spaces for nonlinear, analytically intractable actions. Given a dataset consisting of short tuple/sequences $(x_1, x_2, \ldots)$ that are generated by an unknown group action, NFT conducts Fourier analysis that is composed of (i) learning of an equivariant latent feature on which the group action is *linear*, and (ii) the study of decomposed latent feature as a direct sum of action-equivariant subspaces, which correspond to frequencies. Unlike the previous approaches to equivariance learning that rely on model architectures (Keller and Welling, 2021; Cohen and Welling, 2017), the learning (i) of NFT does not assume any analytically tractable knowledge of the group action in the observation space, and simply uses an autoencoder-type model to infer the actions from the data. In addition to the proposed framework of NFT, we detail our theoretical and empirical contributions as follows.

1. We answer the essential theoretical questions of NFT (Sec 4). In particular,
   - *Existence.* We elucidate when we can find linear equivariant latent features and hence when we can conduct spectral analysis on a generic dataset.
   - *Uniqueness.* We show that NFT associates a nonlinear group action with a set of irreducible representations, assuring NFT's ability to find the unique set of the equivariant subspaces.
   - *Finite-dimensional approximation.* We show that the autoencoder-type loss chooses a set of irreducible representations in approximating the group action in the observation space.
2. We experimentally show that:
   - NFT conducts a data-dependent, nonlinear spectral analysis. It can compress the data under nonlinear deformation and favorably extract the dominant modes of symmetry (Sec 5.1).
   - By using knowledge about the group, NFT can make inferences even when the action is not invertible in the observation space. For example, occlusion can be resolved (Sec 5.2).
   - By introducing prior knowledge of the irreducible representations, we can improve the out-of-domain (OOD) generalization ability of the features extracted by the encoder (Sec 5.2).

## 2    PRELIMINARIES

In this paper, $G$ is a group, and $e$ its unit element. We say $G$ *acts* on a set $\mathcal{X}$ if there is a map $G \times \mathcal{X} \to \mathcal{X}$, $(g, x) \mapsto g \circ x$, such that $e \circ x = x$ and $(gh) \circ x = g \circ (h \circ x)$ for any $x \in \mathcal{X}$ and $g, h \in G$. When $G$ acts on $\mathcal{X}$ and $\mathcal{Y}$, a map $\Phi : \mathcal{X} \to \mathcal{Y}$ is called *equivariant* if $\Phi(g \circ x) = g \circ \Phi(x)$

for any $g \in G, x \in \mathcal{X}$. A *group representation* is defined as a linear action of group $G$ on some vector space $V$, i.e., a group homomorphism $\rho : G \to GL(V)$. See also Appendix A for notations.

**Fourier transform as an equivariant map.** We first explain the equivariance of the classic Discrete Fourier Transform (DFT) to motivate the model and learning of NFT. As is well known, DFT on the function space $L_2^T := \{f : \mathbb{Z}_T \to \mathbb{C}\}$, where $\mathbb{Z}_T = \{j/T \in [0,1) \mid j = 0, \ldots, T-1\}$ (mod $T$) is the $T$ grid points on the unit interval. DFT and its inverse (IDFT) are given by

$$\hat{f}_k = \tfrac{1}{\sqrt{T}} \sum_{j=0}^{T-1} e^{-2\pi i \frac{kj}{T}} f(j/T), \qquad f(j/T) = \tfrac{1}{\sqrt{T}} \sum_{k=0}^{T-1} e^{2\pi i \frac{kj}{T}} \hat{f}_k \quad \forall k, j \in \mathbb{Z}_T. \qquad (1)$$

We can define the group of shifts $G := \mathbb{Z}_T$ (mod $T$) acting on $\mathbf{f} = (f(j/T))_{j=0}^{T-1}$ by $m \circ \mathbf{f} := (f((j-m)/T))_{j=0}^{T-1}$. With the notation $\Phi_k(\mathbf{f}) := \hat{f}_k$, it is well known that $\Phi_k(m \circ \mathbf{f}) = e^{2\pi i \frac{mk}{T}} \Phi_k(\mathbf{f})$ for all $m, k \in \mathbb{Z}_T$, establishing DFT $\Phi$ as an equivariant map; namely

$$\Phi(m \circ \mathbf{f}) = D(m)\Phi(\mathbf{f}) \quad \text{or} \quad m \circ \mathbf{f} = \Phi^{-1}\left(D(m)\Phi(\mathbf{f})\right), \qquad (2)$$

where $D(m) := \mathrm{Diag}(e^{2\pi i \frac{mk}{T}})_{k=0}^{T-1}$ is a diagonal matrix. By definition, $D(m)$ satisfies $D(m+m') = D(m)D(m')$, meaning that $G \ni m \mapsto D(m) \in GL(\mathbb{C}^T)$ is a group representation. Thus, the DFT map $\Phi$ is an equivariant linear encoding of $L_2^T$ into the direct sum of eigen spaces (or the spaces that are invariant with respect to the shift actions), and $\Phi^{-1}$ is the corresponding linear decoder.

In group representation theory, the diagonal element $e^{2\pi i \frac{mk}{T}}$ of $D(m)$ is known as an *irreducible representation*, which is a general notion of *frequency*. **We shall therefore use the word *frequency* and the word *irreducible representation* interchangeably in this paper**. A group representation of many groups can be decomposed into a direct sum of irreducible representations, or the finest unit of group invariant vector spaces. Generally, for a locally compact group $G$, the Fourier transform, the inversion formula, and the frequencies are all analogously defined (Rudin, 1991). For the group action $(g \circ f)(x) = f(g^{-1}x)$, an equivariant formula analogous to eq.(2) also holds with $D(g)$ being a block diagonal matrix. Thus, the frequencies are not necessarily scalar-valued, but matrix-valued.

## 3 NEURAL FOURIER TRANSFORM

In NFT, we assume that a group $G$ acts on a generic set $\mathcal{X}$, and examples of $(x, g \circ x)$ $(x \in \mathcal{X}, g \in G)$ can be observed as data. However, we do not know how the action $\circ$ is given in $\mathcal{X}$, and the action can only be inferred from the data tuples. As in FT, the basic framework of NFT involves an encoder and a decoder, which are to be learned from a dataset to best satisfy the relations analogous to eq.(2):

$$\Phi(g \circ x) = M(g)\Phi(x) \quad \text{and} \quad \Psi(\Phi(x)) = x \quad (\forall x \in \mathcal{X}, \forall g \in G), \qquad (3)$$

where $M(g)$ is *some* **linear map** dependent on $g$, which might be either known or unknown. It turns out that realizing eq.(3) is enough to guarantee that $M(g)$ is a group representation:

**Lemma 3.1.** *If* $span\{\Phi(\mathcal{X})\}$ *is equal to the entire latent space, then eq.(3) implies that $M(g)$ is a group representation, that is, $M(e) = Id$ and $M(gh) = M(g)M(h)$.*

The proof is given in Appendix B. The encoder $\Phi$ and decoder $\Psi$ may be chosen without any restrictions on the architecture. In fact, we will experiment NFT with MLP/CNN/Transformer whose architectures have no relation to the ground truth action.

Given a pair $(\Phi, \Psi)$ that satisfies eq.(3), we also seek an invertible linear map $P$ to block-diagonalize $M(g)$ unless we know $M(g)$ *a priori* in a block-diagonal form. This corresponds to *irreducible decomposition* in representation theory. Assuming that the representation is *completely reducible*, we can seek a common change of basis matrix $P$ for which $B(g) = PM(g)P^{-1}$ is block-diagonal for every $g$ so that each block corresponds to an irreducible component of $M(g)$. See Sec A for the details on irreducible decomposition. Putting all together, NFT establishes the relation

$$g \circ x = \Psi\left(P^{-1}B(g)P\Phi(x)\right). \qquad (4)$$

We call the framework consisting of eqs.(3) and (4) *Neural Fourier Transform (NFT)*, where $z = P\Phi(x)$ is the Fourier image of $x$, and $\Psi(P^{-1}z)$ is the inverse Fourier image. See also Fig 2 for the visualization of the NFT framework. The classic DFT described in Sec 2 is an instance of NFT; $\mathcal{X}$ is $\mathbb{R}^T$, which is the function space on $G = \mathbb{Z}_T$ with shift actions, and $P\Phi$ and $\Psi P^{-1}$ are respectively DFT and IDFT (linear). (Miyato et al., 2022) emerges as an implementation of NFT when $(\Phi, \Psi)$ is to be learned in a completely unsupervised manner with no prior knowledge of $M(g)$ nor $G$ itself. As we show next, however, NFT can be conducted in other situations as well.

## 3.1 THREE TYPICAL SCENARIOS OF NFT

There can be various setting of data and prior knowledge of $G$, and accordingly various methods for obtaining a pair $(\Phi, \Psi)$ in eq.(3). However, we at least need a dataset consisting of tuples of observations (e.g $(x, g \circ x)$) from which we need to infer the effect of group action. A common strategy is to optimize $\mathbb{E}[\|g \circ X - \Psi(M(g)\Phi(X))\|^2]$ where $M(g)$ is either estimated or provided, depending on the level of prior knowledge on $g$ or $G$.

**Unsupervised NFT (U-NFT): Neither $G$ nor $g$ is known.** In U-NFT, the dataset consists of tuples of observations $\{x^{(i)} := (x_0^{(i)}, \ldots, x_T^{(i)})\}_{i=1}^N$, where each $x_t^{(i)}$ is implicitly generated as $x^{(i)} = g_i^t \circ x_0^{(i)}$ for some unobserved $g_i$ sampled from $G$ that is unknown. Such a dataset may be obtained by collecting short consecutive frames of movies/time-series, for example. (Miyato et al., 2022) is a method for U-NFT. MSP uses a dataset consisting of consecutive triplets ($T = 2$), such as any consecutive triplet of images in Fig 1. Given such a dataset, MSP trains $(\Phi, \Psi)$ by minimizing

$$\mathbb{E}[\|x_2 - \Psi(M^*(\Phi(x_1))\|^2], \quad \text{where } M^* = \arg\min_M \|\Phi(x_1) - M\Phi(x_0)\|^2 \tag{5}$$

is computed for each triplet $x$ (Fig 20, Appendix). By considering $\Phi$ with matrix output of dimension $\mathbb{R}^{a \times m}$, $M^* \in \mathbb{R}^{a \times a}$ can be analytically solved as $\Phi(x_1)\Phi(x_0)^\dagger$ where $A^\dagger$ is the Moore Penrose inverse of $A$ (Inner optimization part). Thus, $(\Phi, \Psi)$ is trained in an end-to-end manner by minimizing $E[\|x_2 - \Psi(\Phi(x_1)\Phi(x_0)^\dagger(\Phi(x_1))\|^2]$. For the U-NFT experiments, we used MSP as a method of choice. After training $(\Phi, \Psi)$, we may obtain $M^*$ for each $x^{(i)}$ (say $M_i^*$), and use (Maehara and Murota, 2011) for example to search for $P$ that simultaneously diagonalizes all $M_i^*$s.

$G$**-supervised NFT (G-NFT): $G$ is known but not $g$.** G-NFT has the same dataset assumption as U-NFT, but the user is allowed to know the group $G$ from which each $g_i$ is sampled. In this case, we can assume that the matrix $M(g)$ in the latent space would be a direct sum of irreducible representations of $G$. For example, we may assume some parametric form of irreducible representations $M(\theta) = \oplus_k M_k(\theta)$ and estimate $\theta^{(i)}$ for every tuple of data $x^{(i)}$. However, estimating $\theta^{(i)}$ for each $i$ may not scale for a large dataset. Alternatively, we may just use the dimensional information of the matrix decomposition and estimate each block in the same manner as U-NFT. For instance, in the context of Fig 1, the user may know that the transformation between consecutive frames is "some" periodic action (cyclic shift), for which it is guaranteed that the matrix representation is a direct sum of $2 \times 2$ matrices. When $T = 2$, we can minimize the same prediction loss as in eq.(5) except that we put $M^* = \bigoplus_k^{a/2} M_k^* \in \mathbb{R}^{a \times a}$ where $M_k^* = \arg\min_M \|\Phi_k(x_1) - M\Phi_k(x_0)\|^2 \in \mathbb{R}^{2 \times 2}$ and $\Phi_k$s are the matrices constituting $\Phi$ by vertical concatenation $\Phi = [\Phi_1; \Phi_2; ...]$.

$g$**-supervised NFT (g-NFT): $g$ is known.** In g-NFT, the user can observe a set of $(x, g \circ x)$ for known $g$ so that the data technically consists of $((x, g \circ x), g)$. In the context of Fig 1, the g-NFT setting not only allows the user to know that $g$ is periodic, but also the velocity of the shift (e.g., the size of the pixel-level shift before applying the fisheye transform). Thus, by deciding the irreducible representations to use in our approximation of the action, we can predetermine the explicit form of $M(g)$. For g-NFT, we may train $(\Phi, \Psi)$ by minimizing

$$E[\|g \circ x - \Psi(M(g)\Phi(x))\|^2] + E[\|\Phi(g \circ x) - M(g)\Phi(x)\|^2].$$

The matrix $M(g)$ can be derived from the knowledge of representation theory. For example, if $\mathbb{Z}_N$ is the group, $M(g)$ corresponding to the frequencies $\{f_1, .. f_n\}$ would be the direct sum of the 2D rotation matrices by angle $2\pi f_k g / N$ (see also Fig. 21 in Appendix).

## 3.2 PROPERTIES OF NFT

By realizing eq.(3) approximately, NFT learns spectral information from the data and actions (Lemma 3.1). Here, we emphasize three practically important properties of NFT. First, by virtue of the nonlinear encoder and decoder, NFT achieves nonlinear spectral analysis for arbitrary data types. Second, NFT performs data-dependent spectral analysis; it provides decomposed representations only through the frequencies necessary to describe the symmetry in the data. These two properties contrast with the standard FT, where the pre-fixed frequencies are used for expanding functions. Third, the NFT framework has the flexibility to include spectral knowledge about the group in the latent space, as in G-NFT and g-NFT. This further enables us to extract effective features for various tasks. These points will be verified through theory and experiments in the sequel.

## 4 THEORY

**Existence and uniqueness**: Because the goal of NFT is to express the data space in the form of latent linear equivariant subspaces, it is a fundamental question to answer whether this goal is achievable theoretically. Additionally, it is important to ask if the linear action can be expressed by a unique set of irreducible representations. The following Thm 4.1 answers these questions in the affirmative using the notion of group invariant kernel. Let group $G$ act on the space $\mathcal{X}$. A positive definite kernel $k : \mathcal{X} \times \mathcal{X} \to \mathbb{R}$ is called *G-invariant* if $k(g \circ x, g \circ x') = k(x, x')$ for any $x, x' \in \mathcal{X}$ and $g \in G$.

**Theorem 4.1.** *(Existence, uniqueness, and invariant kernel) Let $G$ be a compact group acting on $\mathcal{X}$. There exists a vector space $V$ and a non-trivial equivariant map $\Phi : \mathcal{X} \to V$ if and only if there exists a non-trivial $G$-invariant kernel on $\mathcal{X}$. Moreover, the set of $G$-invariant kernels and the set of equivariant maps to a vector space are in one-to-one correspondence up to a $G$-isomorphism.*

We provide the proof in Appendix C. The implication of Thm 4.1 is twofold. First, Thm 4.1 guarantees the existence of an equivariant map $\Phi$ by the existence of an invariant kernel. In the proof, $\Phi$ emerges as an embedding into the associated reproducing kernel Hilbert space (RKHS). Thus, under mild conditions, there is a latent space that is capable of linearly representing a sufficiently rich group action, since one can easily construct a $G$-invariant kernel with infinite-dimensional RKHS which is dense in $L^2$ space(Thm C.1.1).

Second, Thm 4.1 implies the identifiability. When there are two invertible equivariant maps $\Phi$ and $\tilde{\Phi}$, Thm 4.1 guarantees that there is a pair of corresponding kernels, and we can induce from them a $G$-isomorphism (Sec A) between the corresponding RKHSs. In other words, $\Phi$ and $\tilde{\Phi}$ are connected via $G$-isomorphism, corresponding to the same set of irreducible representations. *This way, we may associate any linearlizable group action to a unique set of irreducible representations.* Similarly, any two solutions for the g-NFT with the *similar* $M(g)$s would also differ only by $G$-isomorphism.

**Finite dimensional Approximation:** When the output dimension of $\Phi$ is small, the space may not fit all members of irreducible representations corresponding to the target group action and it may not be possible to invertibly encode the action and observations as done in the standard FT. In such a case, the expression $x_1 \to \Psi(M(g)\Phi(x_1))$ in NFT would only provide an approximation of the transition $x_1 \to x_2$. However, what type of approximation would it be in U-NFT and G-NFT? Below we will present the claim guaranteeing that NFT conducts a data-dependent filter that selectively extracts the dominant spectral information in the dataset describing the action. We present an informal statement here (see Sec C.3.2 for the formal version).

**Theorem 4.2.** *(Nonlinear Fourier Filter) Suppose that there exists an invariant kernel $k$ for the group action on $\mathcal{X}$ whose RKHS is large enough to invertibly encode $\mathcal{X}$. Suppose further that $(\Phi^*, \Psi^*)$ is the minimizer of*

$$\mathbb{E}_{g \in G}[\|g \circ X - \Psi\Phi(g \circ X)\|^2]$$

*among the set of all equivariant autoencoders of **a fixed latent dimension** for which the actions are linear in the latent space. Then the frequencies to appear in the latent space of $\Phi^*$ are determined by the signal strength of each irreducible component in the RKHS of $k$.*

This result claims that, in the application of NFT with small latent dimension, U-NFT and G-NFT automatically select the set of irreducible representations that are dominant in describing the action in the observation space, functioning as a data-dependent nonlinear *filter* on the dataset. Please also see Sec C for the theory of NFT. As we will validate experimentally in Sec 5.1, NFT *does* choose the major frequencies discretely even in the presence of noise frequencies.

## 5 EXPERIMENTS

### 5.1 NFT VS DFT

To verify that NFT performs nonlinear spectral analysis, we conduct experiments on 1D time series with time-warped shifts. We first prepared a fixed or random set of frequencies $F$ and constructed

$$\mathbb{Z}_N \ni t \mapsto \ell(t) = \sum_{k=1}^{K} c_k \cos(2\pi f_k (t/N)^3) := r((t/N)^3), \qquad F = \{f_1, f_2, ...., f_K\},$$

which is the time-warped version of the series $r(\cdot) = \sum_{k=1:K} c_k \cos(2\pi f_k \cdot /N)$. The shift $m \in \mathbb{Z}_N$ acts on the latent space as $\ell(\cdot)$ by $(m \circ \ell)(t) = r((t/N)^3 - m/N)$. We used $N = 128$. Note that the frequency spectrum obtained by the classical DFT does not reflect the true symmetry structure defined with $F$ (Fig 3). Codes are in supplementary material, details are in Appendix D.

**Spectral analysis of time-warped signals.** In this experiment, we verify that NFT recovers the fixed frequencies $F$ from the time-warped signals. We generated 30000 sets of $\{c_1, ...c_K\}$ with $K = 7$, yielding 30000 instances of $\ell$ as dataset $\mathcal{X} \subset \mathbb{R}^{128}$. The $c_k$s were sampled from the uniform distribution, and $c_5, c_6$ scaled by the factor of $0.1$ to produce two *noise* frequencies. See Fig 4 for visualization. To train NFT, we prepared a set of sequences of length-3 ($\mathbf{s} = (\ell_0, \ell_1, \ell_2)$ with $\ell_k = r((t/N)^3 - kv_\ell/N) \in \mathbb{R}^{128})$ shifting with random velocity $v_\ell$. We then conducted U-NFT as in Sec 3.1 with latent dimension $\mathbb{R}^{10 \times 16}$ so that for each sequence $\mathbf{s}$,

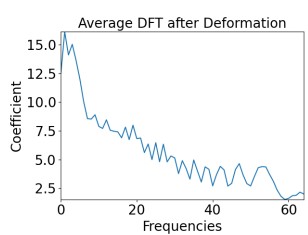

Figure 3: DFT result

the matrix $M_* \in \mathbb{R}^{10 \times 10}$ provides $\ell_t \approx \Psi(M_*^t \Phi(\ell_0))$. With this setting, $\Phi$ can express at most $10/2 = 5$ frequencies.

To check whether the frequencies $F$ obtained by NFT are correct, we use the result from the representation theory stating that, if $\rho_f : \mathbb{Z}_N \to \mathbb{R}^{d_f \times d_f}$ is the irreducible representation corresponding to the frequency $f$, the character inner product (Fulton and Harris, 1991) satisfies

$$\langle \rho_f | \rho_{f'} \rangle = \tfrac{1}{N} \sum_{g \in \mathbb{Z}_N} \text{trace}(\rho_f(g))\text{trace}(\rho_{f'}(g)) = \delta_{ff'} \qquad (\delta_{ff'} \text{ is Kronecker's delta}).$$

We can thus validate the frequencies captured in the simultaneously block-diagonalized $M_*$s by taking the character inner product of each identified block $B_i$ with the known frequencies. The center plot in Fig 5 is the graph of $\mathbb{E}[\langle \rho_f | B_i \rangle]$ plotted agasint $f$. We see that the spike only occurs at the major frequencies in $F$ (i.e. $\{8, 15, 22, 40, 45\}$), indicating that our NFT successfully captures the major symmetry modes hidden in the dataset without any supervision. When we repeated this experiment with 100 instances of randomly sampled $F$ of the same size, NFT was able to recover the major members of $F$ with $(FN, FP) = (0.035, 0.022)$ by thresholding $\mathbb{E}[\langle \rho_f | M_* \rangle]$ at $0.5$. This score did not change much even when we reduced the number of samples to 5000 $(0.04, 0.028)$. This experimental result also validates that the disentangled features reported in (Miyato et al., 2022) are in fact the set of major frequencies in the sense of classical DFT. We can also confirm that underlying symmetry can be recovered even when the deformation is applied to 2D images (Fig 6).

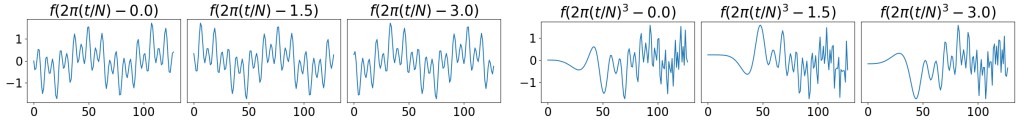

Figure 4: Left:the sequence of length=128 signals constructed by applying the shift operation with constant speed. Right: the sequence of the same function with time deformation.

**Data compression with NFT.** In signal processing, FT is also used as a tool to compress data based on the shift-equivariance structure of signals. We generated 5000 instances of the time-warped signals with $F \subset \{0 : 16\}$, and applied U-, G-, and g-NFT to compress the dataset using encoder output of dimension $\mathbb{R}^{32 \times 1}$. We also added an independent Gaussian noise at each time point in the training signals. As for G-NFT, we used the direct sum of 16 commutative $2 \times 2$ matrices of form $((a, -b), (b, a))$ to parameterize $M^1$. For g-NFT, we used the $2 \times 2$ block diagonal matrix $M(\theta) = \bigoplus_{\ell=0}^{15} \begin{pmatrix} \cos l\theta & -\sin l\theta \\ \sin l\theta & \cos l\theta \end{pmatrix}$ with known $\theta$.

We evaluated the reconstruction losses of these NFT variants on the test signals and compared them against DFT over frequencies of range $\{0 : 16\}$ (Fig 12, Appendix). The mean squared errors together with standard deviations in parentheses are shown in Table 1, which clearly demonstrates that NFT learns the nonlinearity of observation and compresses the data appropriately according to the hidden frequencies. It is also reasonable that g-NFT, which uses the identity of group element $g$

---

[1]When the scalar field is $\mathbb{R}$, $2 \times 2$ would be the smallest irreducible representation instead of $1 \times 1$ in complex numbers.

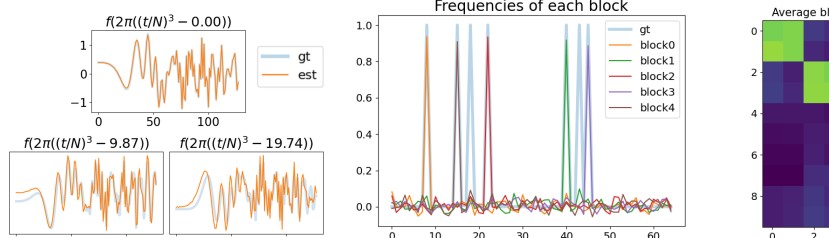

Figure 5: Left: Long horizon future prediction of the sequence of time-warped sigmals. Center: $\mathbb{E}[\langle \rho_f | B \rangle]$ plotted against $f \in [0, 64]$ for each block $B$ in the block diagonalized $M_*$s learned from the dataset with 5 major frequencies ($\{8, 15, 22, 40, 45\}$) and **2 noise frequencies with small coefficients** ($\{18, 43\}$) when $M_*$s can express at most 5 frequencies. Note that $\mathbb{E}[\langle \rho_f | M_* \rangle]$ is linear with respect to $M^*$ (Appendix A.3). Right: Average absolute value of block-diagonalized $M_*$s.

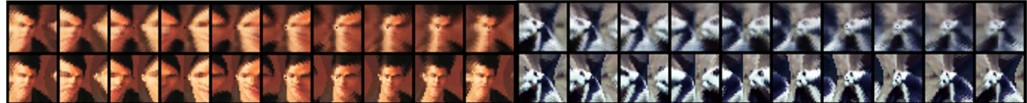

Figure 6: Horizontal shift of unseen objects in fisheye-view predicted from the left-most frame by U-NFT trained on CIFAR100 (Krizhevsky et al., 2009) sequences with $T = 4$. (top:pred, bottom:gt). U-NFT learns the deformed shifts that are not expressible as linear functions on input coordinates.

acting on each signal, achieves more accurate compression than G-NFT. DFT fails to compress the data effectively, which is obvious from Fig 3.

## 5.2 APPLICATION TO IMAGES

We verify the representation ability of NFT by applying it to challenging tasks which involve out-of-domain (OOD) generalization and recovery of occlusion. Codes are in supplementary material.

**Rotated MNIST.** We applied g-NFT to MNIST (LeCun et al., 1998) dataset with SO(2) rotation action and used it as a tool in unsupervised representation learning for OOD generalization. To this end, we trained $(\Phi, \Psi)$ on the in-domain dataset(rotated MNIST), and applied logistic regression on the output of $\Phi$ for the downstream task of classification on the rotated MNIST (in-domain) as well as on rotated Fashion-MNIST (Xiao et al., 2017) and rotated Kuzushiji-MNIST (Clanuwat et al., 2018) (two out-domains). Following the standard procedure as in (Chen et al., 2020), we trained the regression classifier for the downstream task with fixed $\Phi$. We used data consisting of triplets $(g_\theta \circ x, g_{\theta'} \circ x, \theta' - \theta)$ with $x$ being a digit image and $\theta, \theta' \sim \text{Uniform}(0, 2\pi)$ being random rotation angles. We used the same transition matrix $M(\theta)$ used in the data compression experiment (Sec 5.1) with the max frequency $l_{\max} = 2$ plus one-dimensional trivial representation. Details are in Appendix D. Because the representation learned with NFT is decomposed into frequencies, we can make a feature by taking the absolute value of each frequency component; that is, by interpreting the latent variable $\mathbb{R}^{(2l_{\max}+1) \times d_m}$ as $l_{\max} + 1$ frequencies of $d_m$ multiplicity each (trivial representation is 1-dim), we may take the norm of each frequency component to produce $(l_{\max} + 1) \times d_m$-dimensional feature. This is analogous to taking the absolute value of each coefficient in DFT. We designate this approach as **norm** in Fig 7.

As we can see in the right panel of Fig 7, both g-NFT and g-NFT-norm perform competitively compared to conventional methods. In particular, g-NFT norm consistently eclipses all competitors on OOD. In the left panel, although a larger $l_{\max}$ generally benefits the OOD performance, too large a value of $l_{\max}$ seems to result in overfit, just like in the analysis of FT. We also compared NFT against SteerableCNN (Cohen and Welling, 2017), which assumes that acting $G$ is a rotation group. We gave SteerableCNN a competitive advantage by training the model with classification labels on rotMNIST, and fine-tuned the final logit layer on all three datasets, consisting of the in-domain dataset (rotMNIST) and two OOD datasets (rotFMNIST, rotKuzushiji-MNIST). SteerableCNN with supervision outperforms all our baselines in the in-domain dataset, but not on the OOD datasets. We believe that this is because, as pointed out in (Cohen et al., 2018), SteerableCNN functions as a

|  | $g$-NFT | $G$-NFT | DFT ($N_f = 16$) |
|---|---|---|---|
| Noiseless | 3.59 ($\pm$1.29) $\times 10^{-5}$ | 1.98 ($\pm$1.89) $\times 10^{-2}$ | 8.10 |
| $\sigma = 0.01$ | 2.62 ($\pm$0.26) $\times 10^{-4}$ | 2.42 ($\pm$1.19) $\times 10^{-2}$ | – |
| $\sigma = 0.05$ | 1.42 ($\pm$0.14) $\times 10^{-3}$ | 5.82 ($\pm$1.15) $\times 10^{-2}$ | – |
| $\sigma = 0.1$ | 2.53 ($\pm$0.09) $\times 10^{-3}$ | 1.16 ($\pm$0.22) $\times 10^{-1}$ | – |

Table 1: Reconstruction error of data compression for time-warped 1d signals

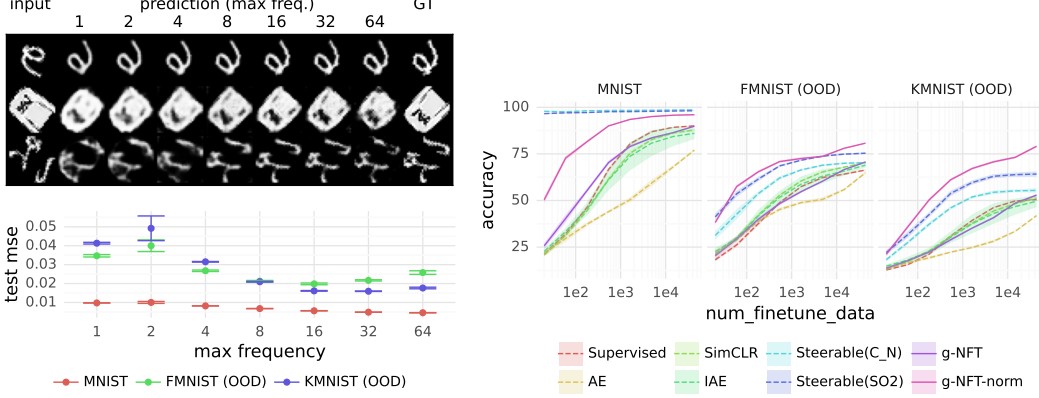

Figure 7: OOD generalization ability of NFT trained on rotMNIST. Top Left: Prediction by g-NFT with various maximum frequencies $l_{\max}$. Bottom Left: Prediction errors of g-NFT. Right: Classification accuracy of linear probe, compared against autoencoder (AE), SimCLR (Chen et al., 2020), the invariant autoencoder (IAE) (Winter et al., 2022) and supervised model including $C_n$ steerable CNN (Cohen and Welling, 2017) and $SO(2)$ steerable CNN (Weiler et al., 2018). Each line is the mean of the accuracy over 10 seeds, with the shaded area being the standard deviation.

composition of filters that preferentially choose the frequencies that are relevant to the task used in the training, so that the model learns G-linear maps that are overfitted to classify the rotMNIST dataset. Also see Appendix E for rotMNIST with more challenging condition involving occlusion.

**Learning 3D Structure from 2D Rendering.** We also applied g-NFT to the novel view synthesis from a single 2D rendering of a 3D object. This is a challenging task because it cannot be formulated as pixel-based transformation — in all renderings, the rear side of the object is always occluded. We used three datasets: ModelNet10-SO3 (Liao et al., 2019) in $64 \times 64$ resolution, BRDFs (Greff et al., 2022) ($224 \times 224$), and ABO-Material (Collins et al., 2022) ($224 \times 224$). Each dataset contains multiple 2D images rendered from different camera positions. We trained the autoencoder with the same procedure as for MNIST, except that we used Wigner D matrix representations of $SO(3)$ with $l_{\max} = 8^2$. We used Vision Transformer (Dosovitskiy et al., 2021) to model $\Phi$ and $\Psi$.

The prediction results (Fig 8) demonstrate that g-NFT accurately predicts the 2D rendering of 3D rotation. We also studied each frequency component by masking out all other frequencies before decoding the latent. Please also see the rendered movie in the Supplementary material. Note that 0-th frequency (F0) captures the features *invariant* to rotation, such as color. F1 (second row) captures the orientation information, and higher frequencies extract symmetries of the object shapes. For example, F3 depicts triangle-like shapes having rotational symmetry of degree 3, similar to the spherical harmonics decomposition done in 3D space (Fig 18). See Appendix D for details.

## 6 RELATED WORKS AND DISCUSSIONS

As a tool in equivariant deep learning, group convolution (GC) has been extensively studied (Cohen and Welling, 2017; Cohen et al., 2019; Krizhevsky et al., 2012). NFT differs from GC as well as FT in that it does not assume $g$ to act linearly on the input. In the words of (Kondor and Trivedi, 2018), FT and GC assume that each instance $x \in \mathcal{X}$ is a function defined on homogeneous space, or the copy of the acting group modulo the stabilizers. These methods must install the explicit knowledge of the group structure into the model architecture (Finzi et al., 2021). (Kondor, 2008; Reisert and

---

[2]We used $SO(2)$ representation for BRDFs however, since its camera positions have a fixed elevation.

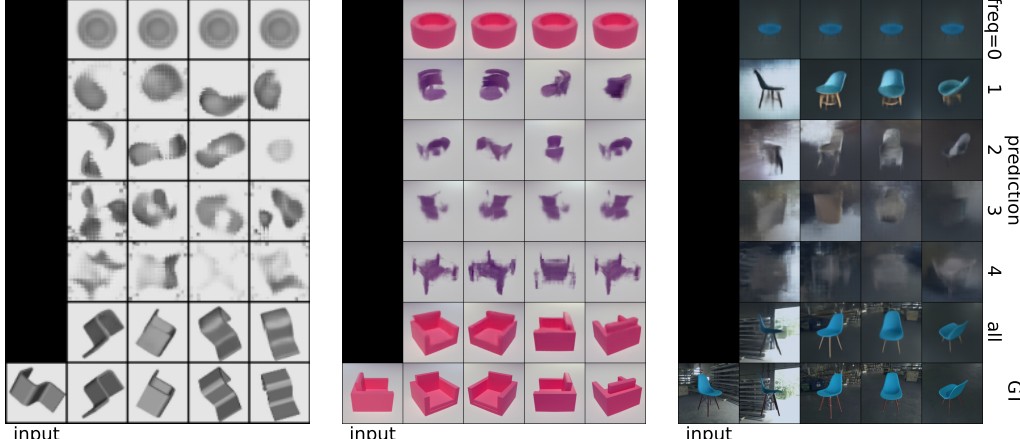

Figure 8: Novel view synthesis from a single image of a test object that is unseen during training (from left to right: ModelNet, BRDFs, ABO). From bottom to top, we show ground truth, g-NFT prediction, and the decoder output with $k$-th frequency only. As for ABO, the same backgrounds were repeatedly used for both training and test datasets.

Burkhardt, 2007) have used group invariant kernel, but limited their scope to situations similar to FT. When the action is linear, our philosophy of G-NFT also shares much in common with (Cohen and Welling, 2014).

As other efforts to find the symmetry under nonlinear actions, (Dupont et al., 2020) took the approach of mapping the input to the latent space of volumetric form and applying the linear rotation operation on the voxels, yielding an instance of g-NFT. (Shakerinava et al., 2022) uses different types of invariants (polynomial) that are specific to group/family of groups, instead of linearized group actions in the form of representations. (Falorsi et al., 2018) maps the observations to the matrix group of $SO(3)$ itself. (Park et al., 2022) first maps the input to an intermediate latent space with a blackbox function and then applies the convolution of known symmetry for contrastive learning. Finally, Koopman operator (Brunton et al., 2022) is closely related to the philosophy of NFT in that it linearizes a *single* nonlinear dynamic, but NFT differs in that it seeks the latent linear transitions with structures (e.g frequencies) that are specific to *group*.

Most relevant to this work, (Miyato et al., 2022) presents an implementation of U-NFT and uses it to disentangle the modes of actions. However, they do not present other versions of NFT (g-NFT, G-NFT) and, most importantly, neither provide the theoretical foundations that guarantee the identifiability nor establish the "learning of linearized equivariance" as an extension of Fourier Transform. By formally connecting the Fourier analysis with their results, the current work has shown that the contextual disentanglement that was often analyzed in the framework of probabilistic generative model (Zhang et al., 2020; Kim and Mnih, 2018) or the direct product of groups (Higgins et al., 2018; Yang et al., 2021) may be described in terms of Fourier frequency as well. To our knowledge, we are the first of a kind in providing a formal theory for seeking the linear representations on the dataset with group symmetries defined with nonlinear actions.

## 7 LIMITATIONS AND FUTURE WORK

As stated in Sec 3.1, NFT requires a dataset consisting of (short) tuples because it needs to observe the transition in order to learn the equivariance hidden in nonlinear actions; this might be a practical limitation. Also, although NFT can identify the major frequencies of data-specific symmetry, it does not identifiably preserve the norms in the latent space and hence the size of the coefficient in each frequency, because the scale of $\Phi$ is undecided. Finally, while NFT is a framework for which we have provided theoretical guarantees of existence and identifiability in general cases, the work remains to establish an algorithm that is guaranteed to find the solution to eq.(3). As stated in (Miyato et al., 2022), the proof has not been completed to assure that the AE loss of Sec 3.1 can find the optimal solution, and resolving this problem is a future work of great practical importance.

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

## A ELEMENTS OF GROUP, ACTION, AND GROUP REPRESENTATION

In this first section of the Appendix, we provide a set of preliminary definitions and terminologies used throughout the main paper and the Appendix section (Sec C) dedicated to the proofs of our main results. For a general reference on the topics discussed here, see (Fulton and Harris, 1991), for example.

### A.1 GROUP AND GROUP REPRESENTATION

A *group* is a set $G$ with a binary operation $G \times G \ni (g, h) \mapsto gh \in G$ such that (i) the operation is associative $(gh)r = g(hr)$, (ii) there is a unit element $e \in G$ so that $eg = ge = g$ for any $g \in G$, and (iii) for any $g \in G$ there is an inverse $g^{-1}$ so that $gg^{-1} = g^{-1}g = e$.

Let $G$ and $H$ be groups. A map $\varphi : G \to H$ is called *homomorphism* if $\varphi(st) = \varphi(s)\varphi(t)$ for any $s, t \in G$. If a homomorphism $\varphi : G \to H$ is a bijection, it is called *isomorphism*.

Let $V$ be a vector space. The set of invertible linear transforms of $V$, denoted by $GL(V)$, is a group where the multiplication is defined by composition. When $V = \mathbb{R}^n$, $GL(\mathbb{R}^n)$ is identified with the set of invertible $n \times n$ matrices as a multiplicative group.

For group $G$ with unit element $e$ and set $\mathcal{X}$, a (left) action $L$ of $G$ on $\mathcal{X}$ is a map $L : G \times \mathcal{X} \to \mathcal{X}$, denoted by $L_g(x) := L(g, x)$, such that $L_e(x) = x$ and $L_{gh} = L_g(L_h(x))$. If there is no confusion, $L_g(x)$ is often written by $g \circ x$.

For group $G$ and vector space $V$, a *group representation* $(\rho, V)$ is a group homomorphism $\rho : G \to GL(V)$, that is, $\rho(e) = Id_V$ and $\rho(gh) = \rho(g)\rho(h)$. A group representation is a special type of action; it consists of linear maps on a vector space. Of particular importance, we say $(\rho, V)$ is *unitarizable* if there exists a change of basis on $V$ that renders $\rho(g)$ unitary for all $g \in G$. It is known that any representation of a compact group is unitarizable (Folland, 1994). Through group representations, we can analyze various properties of groups, and there is an extensive mathematical theory on group representations. See (Fulton and Harris, 1991), for example.

Let $(\rho, V)$ and $(\tau, W)$ be group representations of $G$. A linear map $f : V \to W$ is called *G-linear map* (or *G-map*) if $f(\rho(g)v) = \tau(g)f(v)$ for any $v \in V$ and $g \in G$, that is, if the diagram in Fig 9 commutes.

$$
\begin{array}{ccc}
V & \xrightarrow{\rho(g)} & V \\
f \downarrow & & \downarrow f \\
W & \xrightarrow{\tau(g)} & W
\end{array}
$$

Figure 9: $G$-linear map

A $G$-map is a homomorphism of the representations of $G$. If there is a bijective $G$-map between two representations of $G$, they are called *G-isomorphic*, or *isomorphic* for short.

Related to this paper, an important device in representation theory is irreducible decomposition. A group representation $(\rho, V)$ is *reducible* if there is a non-trivial, proper subspace $W$ of $V$ such that $W$ is invariant to $G$, that is, $\rho(g)W \subset W$ holds for any $g \in G$. If $\rho$ is not reducible, it is called *irreducible*.

There is a famous, important lemma about a $G$-linear map between two *irreducible* representations.

**Theorem A.1** (Schur's lemma). *Let $(V, \rho)$ and $(W, \tau)$ be two irreducible representations of a group $G$. If $f : V \to W$ is G-linear, then $f$ is either an isomorphism or the zero map.*

Figure 10: Simultaneous block diagonalization. Existence of such $P$ is guaranteed if the representation is completely reducible. Note that the matrix $P$ for the change of basis is common for all $g \in G$.

### A.2 IRREDUCIBLE DECOMPOSITION

A representation $(\rho, V)$ is called *decomposable* if it is isomorphic to a direct sum of two non-trivial representations $(\rho_U, U)$ and $(\rho_W, W)$, that is

$$\rho \cong \rho_U \oplus \rho_W, \tag{6}$$

If a representation is not decomposable, we say that it is *indecomposable*.

It is obvious that if a group representation is decomposable, it is reducible. In other words, an irreducible representation is indecomposable. An indecomposable representation may not be irreducible in general. It is known (Maschke's theorem) that for a finite group, this is true; for a non-trivial subrepresentation $(\tau, U)$ of $(\rho, V)$ we can always find a complementary subspace $W$ such that $V = U \oplus W$ and $W$ is $G$-invariant.

It is desirable if we can express a representation $\rho$ as a direct sum of irreducible representations. We say that $(\rho, V)$ is *completely reducible* if there is a finite number of irreducible representations $\{(\rho_i, V_i)\}_{i=1}^m$ of $G$ such that

$$\rho \cong \rho_1 \oplus \cdots \oplus \rho_m. \tag{7}$$

A completely reducible representation is also called *semi-simple*. If the irreducible components are identified with a subrepresentation of the original $(\rho, V)$, the component $\rho_i$ can be uniquely obtained by a restriction of $\rho$ to the subspace $V_i$. The irreducible decomposition is thus often expressed by a decomposition of $V$, such as

$$V = V_1 \oplus \cdots \oplus V_m \tag{8}$$

In the irreducible decomposition, there may be some isomorphic components among $V_i$. By summarizing the isomorphic classes, we can have the *isotypic decomposition*

$$V \cong W_1^{n_1} \oplus \cdots \oplus W_k^{n_k}, \tag{9}$$

where $W_1, \ldots, W_k$ are mutually non-isomorphic irreducible representations of $G$, and $n_j$ is the multiplicity of the irreducible representation $W_j$. The isotypic decomposition is unique if an irreducible decomposition exists.

A group representation may not be completely reducible in general. However, some classes are known to be completely reducible. For example,

- a representation of a finite group
- a finite-dimensional representation of a compact Lie group
- a finite-dimensional representation of a locally compact Abelian group

are completely reducible (Fulton and Harris, 1991).

If a representation $(\rho, V)$ is completely reducible, we can find a basis of $V$ such that the representation can be expressed with the basis in the form of block diagonal components, where each block corresponds to an irreducible component. Note that the basis does not depend on group element $g$, and thus we can realize simultaneous block diagonalization (see Fig 10).

When complex vector spaces are considered, any irreducible representation of a locally compact Abelian group is one-dimensional. This is the basis of Fourier analysis. For the additive group $[0, 1)$, they are the Fourier basis functions $e^{2\pi k i x}$ ($x \in [0, 1)$, $k = 0, 1, \ldots$). See (Rudin, 1991), for example, for Fourier analysis on locally compact Abelian groups.

A.3 CHARACTERS

In this section, we introduce the idea of characters which is used in Sec 5.1. For a representation $(\rho, V)$, the character $\chi_\rho : G \to \mathbb{C}$ of $\rho$ is defined as its trace

$$\chi_\rho(g) = \text{trace}(\rho(g))$$

This function has extremely useful properties in harmonic analysis, and we list two of them that we exploit in this paper.

**Property 1 (Invariance to change of basis)** By its definition, character is invariant to change of basis. That is, if $(\rho, V)$ and $(\tilde{\rho}, W)$ are two representations that are isomorphic so that there exists a linear invertible map $P : V \to W$, then

$$\chi_{\tilde{\rho}(g)} = \text{trace}(\tilde{\rho}(g)) \tag{10}$$
$$= \text{trace}(P^{-1}\rho(g)P) \tag{11}$$
$$= \text{trace}(\rho(g)) = \chi_\rho(g). \tag{12}$$

Indeed, if $V = W$, $P$ is just the change of basis.

**Property 2 (Character orthogonality)** Suppose that $G$ is a compact group. Then group invariant measure on $G$ is a measure satisfying that

$$\mu(A) = \mu(gA) := \mu(\{g \circ a; a \in A\}).$$

If $d\mu$ is a group invariant measure with $\int_G d\mu(g) = 1$, then for all irreducible representations $\rho$ and $\tilde{\rho}$,

$$\langle \rho | \tilde{\rho} \rangle = \int_G \overline{\chi_\rho(g)} \chi_{\tilde{\rho}}(g) d\mu(g) = \begin{cases} 1 & \text{if } \rho \text{ and } \tilde{\rho} \text{ are isomorpohic} \\ 0 & \text{otherwise.} \end{cases} \tag{13}$$

Thus, if $M$ is a representation that is isomorphic to the direct sum of $\{\rho_k\}$, then by the linear property of the trace, $\langle \rho | M \rangle$ is the number of components of the direct sum that are isomorphic to $\rho$, and this is called *multiplicity of $\rho$ in $M$*. For more detailed theory of characters, see (Fulton and Harris, 1991).

## B   PROOF OF LEMMA 3.1

The following is a rephrase of Lemma 3.1.

**Lemma B.1.** *Suppose that*
$$\Phi(g \circ x) = M(g)\Phi(x) \tag{14}$$
*holds for any $x \in \mathcal{X}$ and $g \in G$. If $\text{span}\{\Phi(\mathcal{X})\}$ equals the entire latent space, then $M(g)$ is a group representation, i.e., $M(e) = Id$ and $M(gh) = M(g)M(h)$.*

*Proof.* Take $g = e$ in eq.(14). Then, $\Phi(x) = \Phi(e \circ x) = M(e)\Phi(x)$ for any $x$. It follows from the assumption of $\{\Phi(\mathcal{X})\}$ that $M(e) = Id$ holds. Next, by replacing $x$ with $h \circ x$ ($h \in G$) in eq.(14), we have

$$\Phi(g \circ (h \circ x)) = M(g)\Phi(h \circ x).$$

This implies

$$M(gh)\Phi(x) = M(g)M(h)\Phi(x),$$

from which we have $M(gh) = M(g)M(h)$. □

## C   PROOFS OF THEOREMS IN SEC 4

This section gives the proofs of the theorems in Sec 4. Throughout this section, we use $\mathcal{X}$ to denote a data space, and assume that a group $G$ acts on $\mathcal{X}$.

In this paper, we discuss an extension of Fourier transform to any data. Unlike the standard Fourier transform, it does not assume each data instance to be a function on some homogeneous space. To

this end, we embed each point $x \in \mathcal{X}$ to a function space on $\mathcal{X}$, on which a linear action or group representation is easily defined. Recall that we can define a *regular representation* on a function space $\mathcal{H}$ on $\mathcal{X}$ by

$$L_g : f(\cdot) \mapsto f(g^{-1}\cdot) \qquad (g \in G).$$

It is easy to see that $g \mapsto L_g$ is a group representation of $G$ on the vector space $\mathcal{H}$. In order to discuss the existence of $\Phi$ into a latent space with linear action, we want to introduce an embedding $\Phi : \mathcal{X} \to \mathcal{H}$ and make use of this regular representation. The reproducing kernel Hilbert space is a natural device to discuss such an embedding, because, as is well known to the machine learning community, we can easily introduce the so-called *feature map* for such a $\Phi$, which is a mapping from space $\mathcal{X}$ to the reproducing kernel Hilbert space.

## C.1 EXISTENCE

In this subsection, we will prove the existence part of Thm 4.1; that is, whenever there is a feature space with $G$-invariant inner product structure, we can find a linear representation space for the data space, which means that we can find an encoding space where the group action is linear.

**RKHS with $G$-invariant kernel:**

**Proposition C.1.** *Suppose that $K$ is a positive definite kernel on a topological space on $\mathcal{X}$ and $G$ acts continuously on $\mathcal{X}$. Define $K_g(x,y) := K(g^{-1}x, \ g^{-1}y)$ for $g \in G$. Then $H(K_g) := \{f(g^{-1}\cdot); f \in H(K)\}$ is the RKHS corresponding to $K_g$ for any $g$, and $H(K) \to H(K_g)$ defined by $\rho(g) : f(\cdot) \mapsto f(g^{-1}\cdot)$ is an isomorphism of Hilbert spaces.*

*Proof.* It is clear that $K_g(x,y)$ is positive definite and that $\rho(g)$ is an invertible linear map. Let $H_0$ be the dense linear subspace spanned by $\{K_g(\cdot, x)\}_x$ in $H(K_g)$. In the way of Moore's theorem, we equip $H_0$ with the inner product via $\langle k_g(x, \cdot), k_g(y, \cdot)\rangle_{K_g} = k_g(x, y)$. First, note that $K_g$ satisfies the reproducing property for any function $f \in H_0$. In fact, for $f = \sum_n a_n K_g(\cdot, x_n)$, we see

$$\langle f, K_g(\cdot, x)\rangle_{K_g} = \sum_n a_n K_g(x_n, x) = f(x).$$

Also, $H(K_g)$ is the closure of the span of $\{k_g(x, \cdot)\}$ with respect to the norm defined by this inner product. It is then sufficient to show that $\rho(g)$ maps the linear span of $\{K(\cdot, x)\}_x$ to $H_0$ isometrically. Consider $f_N(\cdot) = \sum_{n=1}^N a_n k(x_n, \cdot) \in H$ and note that

$$\rho(g)(f_N) = f_N(g^{-1}\cdot) = \sum_{n=1}^N a_n k(x_n, g^{-1}\cdot) \tag{15}$$

$$= \sum_n^N a_n k_g(gx_n, \cdot) \in H_0. \tag{16}$$

From this relation, we see

$$\begin{aligned} \|\rho(g)(f_N)\|_{K_g}^2 &= \sum a_n a_m k_g(gx_n, gx_m) \\ &= \sum a_n a_m k(x_n, x_m) \\ &= \|f_N\|_K^2. \end{aligned} \tag{17}$$

This completes the proof. $\qquad\square$

Now, this result yields the following important construction of a representation space when $K$ is $G$-invariant.

**Corollary C.2.** *Suppose that $K$ is a positive definite kernel on $\mathcal{X}$ that is $G$-invariant in the sense that $K(g^{-1}x, g^{-1}y) = K(x, y)$ for any $x, y \in \mathcal{X}$ and $g \in G$. Then the reproducing kernel Hilbert space $H(K)$ is closed under the linear representation of $G$ defined by*

$$\rho(g) : f(\cdot) \to f(g^{-1}\cdot),$$

*and $\rho$ is unitary.*

*Proof.* If $K_g = K$, then $H(K_g) = H(K)$ for any $g$, and thus $\rho(g)$ acts on $H(K)$ linearly. Also, it follows from eq.(17) in the proof of Proposition C.1 that $\|h\|_K^2 = \|h(g^{-1}\cdot)\|_K^2$ for $h \in H(K)$. Therefore $\rho(g)$ is unitary. $\qquad\square$

The following is just a restatement of the corollary C.2.

**Corollary C.3.** *Let $\mathcal{X}$ be a set for which $G$-invariant positive definite kernel $K$ exists. Then there exists a latent vector space $V$ that is a representation space of $G$, i.e., there is an equivariant map $\Phi : \mathcal{X} \to V$ such that the action of $G$ on $V$ is linear. .*

*Proof.* Simply define the map $\Phi : x \to k_x(\cdot) = K(x, \cdot)$. Note that $\rho(g)$ defines an appropriate group representation on the span of $\{k_x(\cdot) = K(x, \cdot)\}$ because $k_{g \circ x}(\cdot) = K(g \circ x, \cdot) = K(x, g^{-1}\cdot) = \rho(g)k_x(\cdot)$ by the invariance property, and the result follows with $V = H(K)$. $\qquad\square$

Thus, this result shows that an equivariant feature can be derived from any invariant kernel.

### C.1.1 THE RICHNESS OF THE SPACE GENERATED BY GROUP INVARIANT KERNEL

Given a positive definite kernel $K(x, y)$, the integral with the Haar measure easily defines a $G$-invariant kernel. An important requirement for the $G$-invariant kernel is that it can introduce a sufficiently rich reproducing kernel Hilbert space (RKHS) as a latent vector space so that any irreducible representation is included. The following theorem shows that if the RKHS of the kernel $K$ is dense in $L_2(P)$, then so is the RKHS of the derived $G$-invariant kernel.

**Theorem C.4.** *Let $G$ be a locally compact group acting continuously on a space $\mathcal{X}$, and $K$ be a continuous positive definite kernel on $\mathcal{X}$ such that the corresponding reproducing kernel Hilbert space $\mathcal{H}_K$ is dense in $L_2(P)$ with a probability measure $P$ on $\mathcal{X}$. Then, assuming that, for any $x, y \in \mathcal{X}$, the integral with right Haar measure $\mu$ is bounded $\int_G |K(g \circ x, g \circ y)|d\mu(g) < C$ for a constant $C$, the positive definite kernel defined by*

$$K_G(x, y) := \int_G K(g \circ x, g \circ y)d\mu(g) \tag{18}$$

*is $G$-invariant such that the corresponding reproducing kernel Hilbert space is dense in $L_2(P)$.*

*Proof.* The invariance is easily confirmed since, for any $a \in G$,

$$K_G(a \circ x, a \circ y) = \int_G K(ga \circ x, ga \circ y)d\mu(g) = \int_G K(g \circ x, g \circ y)d\mu(g)$$

holds by the right invariance property of $\mu$.

For the denseness in $L_2(P)$, suppose that $h \in L_2(P)$ is orthogonal to $\mathcal{H}_{K_G}$ in $L_2(P)$. It suffices to show that $h = 0$.

It follows from the orthogonality that

$$\int_{\mathcal{X}} \int_{\mathcal{X}} \int_G K(g \circ x, g \circ y)d\mu_G(g)h(x)h(y)dP(x)dP(y) = 0. \tag{19}$$

Let $\phi(x) := K(\cdot, x) \in \mathcal{H}_K$. From $\int_{\mathcal{X}} \int_{\mathcal{X}} \int_G |K(g \circ x, g \circ y)h(x)h(y)|dP(x)dP(y)d\mu(g) \le C\|h\|_{L_2(P)}^2$, Fubini's theorem tells that the left-hand side of eq.(19) equals to

$$\int_G \int_{\mathcal{X}} \int_{\mathcal{X}} K(g \circ x, g \circ y)h(x)h(y)dP(x)dP(y)d\mu(g)$$

$$= \int_G \int_{\mathcal{X}} \int_{\mathcal{X}} \langle \phi(g \circ x)h(x), \phi(g \circ y)h(y) \rangle_{\mathcal{H}_K} dP(x)dP(y)d\mu(g)$$

$$= \int_G \|M_h(g)\|_{\mathcal{H}_K}^2 d\mu(g), \tag{20}$$

where $M_h(g)$ is defined by

$$M_h(g) := \int_{\mathcal{X}} \phi(g \circ x)h(x)dP(x).$$

Note that $M_h : G \to \mathcal{H}_K$ is well-defined and continuous by the Bochner integral, since

$$\int_{\mathcal{X}} \|\phi(g \circ x) h(x)\|_{\mathcal{H}_K} dP(x)$$

$$\leq \left( \int_{\mathcal{X}} \|\phi(g \circ x)\|_{\mathcal{H}_K}^2 dP(x) \right)^{1/2} \left( \int_{\mathcal{X}} |h(x)|^2 dP(x) \right)^{1/2}$$

$$= \left( \int K(g \cdot x, g \cdot x) dP(x) \right)^{1/2} \|h\|_{L_2(P)}$$

is finite and continuous with respect to $g$.

It follows from eq.(19) and eq.(20) that

$$\int_G \|M_h(g)\|_{\mathcal{H}_K}^2 d\mu(g) = 0$$

holds, which implies $M_h(g) = 0$. In particular, plugging $g = e$, we have

$$\int_{\mathcal{X}} \phi(x) h(x) dP(x) = 0.$$

The denseness of $\mathcal{H}_K$ in $L_2(P)$ implies that the integral operator $h \mapsto \int_{\mathcal{X}} K(x,y) h(y) dP(y)$ is injective, and thus we have $h = 0$, which completes the proof. $\qquad\square$

## C.2 UNIQUENESS

Now that we have shown that any given kernel can induce an equivariant map, we will show the other way around and establish Theorem 4.1.

**Theorem C.5.** *Up to $G$-isomoprhism, the family of $G$-Invariant kernel $K(x,y)$ on $\mathcal{X}$ has one-to-one correspondence with the family of equivariant feature $\Phi : \mathcal{X} \to V$ such that the action on $V$ is unitarizable and $V = \mathrm{Cl}\,\mathrm{Span}\{\Phi(x) \mid x \in \mathcal{X}\}$.*

*Proof.* We have shown in Proposition C.3 that an invariant kernel induces a linear representation space with unitary representation. Note that, in the construction of $\Phi$ in C.3 from the kernel, the $\Phi$ trivially satisfies the relation $\langle \Phi(x), \Phi(y) \rangle = K(x,y)$. Next we show that this correspondence from $H(K)$ to $\Phi$ is in fact *one to one* by reversing this construction and show that a given unitarizable equivariant map $\Phi$ can correspond to a unique class of representations that are all isomorphic to $H(K)$ (isomorphism class of $H(K)$). In particular, we start from a unitarizable equivariant map $\Phi : \mathcal{X} \to V$ to construct $K$, and show that $V$ is isomorphic to $H(K)$ itself as a representation space.

Let $\Phi : \mathcal{X} \to V$ be an equivariant map into a linear representation space $(M, V)$ with $V = \mathrm{Cl}\,\mathrm{Span}\{\Phi(x) \mid x \in \mathcal{X}\}$. We assume WLOG that $M(g)$ is unitary for all $g$ because the assumption guarantees that $M$ can be unitarized with a change of basis, which is a $G$-isomorphic map. Let $K$ be a kernel defined by $K(x,y) = \langle \Phi(x), \Phi(y) \rangle$. This kernel is invariant by construction. We will show that $V$ is $G$-isomorphic to $H(K)$.

To show this, we consider the map from $V$ to $\mathcal{H}(K)$ defined as

$$J : V \to \mathcal{H}(K) \qquad \text{such that} \qquad J(u) = [x \mapsto \langle \Phi(x), u \rangle], \qquad (21)$$

where $J(u)$ is a member of $H(K)$ by the definition of $K$. We claim that $\langle \Phi(g^{-1} \circ x), u \rangle = \langle \Phi(x), g \circ u \rangle$ for all $u$. To see this, note that any $u \in \mathrm{Cl}\,\mathrm{Span}\{\Phi(x) \mid x \in \mathcal{X}\}$ can be rewritten as $u = \sum_i a_i \Phi(x_i)$ for some sequence of $a_i$. Thus

$$\langle \Phi(x), g \circ u \rangle = \langle \Phi(x), \sum_i a_i M(g) \Phi(x_i) \rangle = \langle M^*(g) \Phi(x), \ \sum_i a_i \Phi(x_i) \rangle = \langle \Phi(g^{-1} \circ x), \ u \rangle$$

$$(22)$$

where $M^*(g) = M(g^{-1})$ is the conjugate transpose of $M(g)$. Therefore it follows that

$$J(M(g)u) = \{x \to \langle \Phi(x), M(g)u \rangle\} \tag{23}$$

$$= \{x \to \langle \Phi(g^{-1} \circ x), u \rangle\} \tag{24}$$

$$= g \circ J(u) \tag{25}$$

where the action of $g$ on $H(K)$ is the very unitary representation with $g \circ f(x) = f(g^{-1}x)$ that we defined in Sec C.1. Because $J$ is trivially linear, this shows that $J$ is a $G$-linear map with standard action on $\mathcal{H}(K)$. Also, if $\langle \Phi(x), u \rangle = \langle \Phi(x), v \rangle$ for all $x$, then $u = v$ necessarily if $V = \mathrm{Cl\,Span}\{\Phi(x) \mid x \in \mathcal{X}\}$. Thus, the map $J$ is injective. The map $J$ is trivially surjective as well, because $J\Phi(x) = k_x$ and $\mathcal{H}(K) = \mathrm{Cl\,Span}\{k_x \mid x \in \mathcal{X}\}$. In fact, this is map induces an isometry as well, because

$$\langle J\Phi(x), J\Phi(y) \rangle_K = \langle k_x, k_y \rangle_K := K(x, y) = \langle \Phi(x), \Phi(y) \rangle \tag{26}$$

and thus validating the reproducing property $\langle J\Phi(x), Ju \rangle_K = \langle \Phi(x), u \rangle = Ju(x)$. Finally, this isomorphism holds for any $\Phi$ satisfying $K(x, y) = \langle \Phi(x), \Phi(y) \rangle$. In summary, Any group invariant $K$ corresponds to a unique linear representation space $H(K)$, and any family of unitarizable equivariant map $\Phi$s that are $G$-isomorphic to each other corresponds to a unique $G$-invariant kernel $K$ corresponding to $H(K)$ with $K(x, y) = \langle \Phi(x), \Phi(y) \rangle$. $\square$

When the group of concern is compact, this result establishes the one-to-one correspondence with any equivariant map because all the representations of a compact group are unitarizable (Folland, 1994).

This result also derives the following collorary claimed in Sec 4.

**Corollary C.6.** *Suppose that $\Phi_i : \mathcal{X} \to V_i$ is an invertible equivariant map into a linear representation space $(M_i, V_i)$ for $i = 1, 2$. Then $V_1$ and $V_2$ differ by $G$-isomorphic map.*

*Proof.* By the previous claim, if $K_i(x, y) := \langle \Phi_i(x), \Phi_i(y) \rangle$ then it suffices to show that $\mathcal{H}(K_1)$ and $\mathcal{H}(K_2)$ are $G$-isomorphic. By the invertivility of $\Phi_k$ and the previous result, $L : K_1(x, \cdot) \to K_2(x, \cdot)$ is an invertible map on $K_1(\mathcal{X}, \cdot)$. This trivially induces $G$-isomoprhism between $\mathcal{H}(K_1)$ and $\mathcal{H}(K_2)$ because $L(K_1(g^{-1} \circ x, \cdot)) = K_2(g^{-1} \circ x, \cdot) = g \circ K_2(x, \cdot)$ and $K_i(x, \cdot)$ generates $\mathcal{H}(K_i)$. $\square$

## C.3 Fourier Filter

In this paper, we are building a theoretical framework that utilizes an equivariant encoder $\Phi : \mathcal{X} \to V$ into linear representation space $V$ satisfying

$$\Phi(g \circ X) = D(g)\Phi(X) \quad D : G \to GL(V), \quad D(g)D(h) = D(gh) \forall g, h \in G \tag{27}$$

together with a decoder $\Psi$. In this section, we make a claim regarding what we call "Filtering principle" that describes the *information filtering* that happens in the optimization of the following loss:

$$\mathcal{L}(\Phi, \Psi | \mathcal{P}, \mathcal{P}_G) = \mathbb{E}_{X \sim \mathcal{P}, g \sim \mathcal{P}_G}[\|g \circ X - \Psi\Phi(g \circ X)\|^2] \tag{28}$$

where $\Phi$ is to be chosen from the pool of equivariant encoder and $\Psi$ is chosen from the set of all measurable maps mapping $V$ to $\mathcal{X}$. We consider this loss when both $\mathcal{P}$ and $\mathcal{P}_G$ are distributions over $\mathcal{X}$ and $G$ with full support.

### C.3.1 Linear Fourier Filter

Let us begin from a restrictive case where $\mathcal{X}$ is itself a representation space (inner product vector space) with unitary linear action of $G$ granting the isotypic decomposition

$$\mathcal{X} = \bigoplus_b \mathcal{V}_b \tag{29}$$

where $\mathcal{V}_b$ is the direct sum of all irreducible representations of type $b$. $\mathcal{V}_b$ is called isotypic space of type $b$ (Clausen and Baum, 1993; Ceccherini-Silberstein et al., 2007).

In order to both reflect the actual implementation used in this paper as well as to ease the argument, we consider the family of equivariant maps $\Phi : \mathcal{X} \to W$ that satisfy the following properties:

**Definition C.7.** (Multiplicity-unlimited-Mapping)
Let $\mathcal{X}$ be a representation space of $G$ and $\mathcal{I}_{\mathcal{X}}$ be the set of all irreducible types present in $\mathcal{X}$. Suppose we have a vector space $W$ with a linear action of $G$ and an equivariant map $\Phi : \mathcal{X} \to W$. In this case, we use $\mathcal{I}_\Phi$ to be the set of all irreducible types present in $\Phi(\mathcal{X})$. We say that $\Phi$ is a *multiplicity-unlimited map* if the multiplicity of any $b \in \mathcal{I}_\Phi$ is equal to that of $b$ in $\mathcal{X}$.

Thus, in terms of the characters, if $G$ acts on $\Phi(\mathcal{X})$ with the representation $M_\Phi$ and $G$ acts on $\mathcal{X}$ with the representation $M$ (that is, if $\Phi(g \circ x) = M_\Phi(g)\Phi(x)$ and $g \circ x = M(g)x$ $\$forall x \in \mathcal{X}$) , then $\langle \rho | M \rangle = \langle \rho | M_\Phi \rangle$ whenever $\langle \rho | M \rangle > 0$.

Given an equivariant map $\Phi : \mathcal{X} \to W$, where $W$ is a vector space with linear action, let $C_\Phi(\mathcal{X})$ denote the set of all measurable map from $W$ to $\mathcal{X}$. We also use $P_b$ to be the projection of $\mathcal{X}$ onto $\mathcal{V}_b$, and let $V_b = P_b X$. Because each $\mathcal{V}_b$ is a space that is invariant to the action of $G$, $P_b$ can be shown to be a $G$-linear map (Clausen and Baum, 1993). With these definitions, we then make the following claim:

**Proposition C.8.** *Fix a representation space $\mathcal{X}$, a distribution $\mathcal{P}$ on $\mathcal{X}$, and $\mathcal{P}_G$ on $G$. With the assumptions set forth above, let $\mathcal{M}$ be the set of all multiplicity-unlimited linear equivariant map from $\mathcal{X}$ to some vector space $W$ with a linear action. For any given $\Phi \in \mathcal{M}$, define*

$$L(\Phi) := \min_{\Psi \in \mathcal{C}_\Phi(\mathcal{X})} \mathcal{L}(\Phi, \Psi | \mathcal{P}, \mathcal{P}_G),$$

*where $\mathcal{L}(\Phi, \Psi | \mathcal{P}, \mathcal{P}_G)$ is given by eq.(28). Then*

$$L(\Phi) = \sum_{b \notin \mathcal{I}_\Phi} \mathbb{E}_{\mathcal{P}}[\|V_b\|^2]\mathcal{R}_b(\mathcal{I}_\Phi) \tag{30}$$

*for some set-dependent functions $\mathcal{R}_b$.*

*Proof.* Let $P_b$ be the projection of $\mathcal{X}$ onto $\mathcal{V}_b$, so that

$$\mathcal{L}(\Phi, \Psi | \mathcal{P}, \mathcal{P}_G) = \sum_b \mathbb{E}[\|g \circ V_b - P_b \Psi \Phi(g \circ X)\|^2]$$

$$= \sum_b R_b(\Psi, \Phi | \mathcal{P}, \mathcal{P}_G) \tag{31}$$

where $V_b = P_b X$ and the integrating distributions $(\mathcal{P}, \mathcal{P}_G)$ in the suffix of $\mathbb{E}$ are omitted for brevity. Now, it follows from the definition of conditional expectation that

$$\min_\Psi R_b(\Psi, \Phi | \mathcal{P}, \mathcal{P}_G) = \mathbb{E}[g \circ V_b - \mathbb{E}[g \circ V_b \mid \Phi(g \circ X)]]$$

$$= \mathbb{E}[g \circ V_b - \mathbb{E}[g \circ V_b \mid \{g \circ V_k; k \in \mathcal{I}_\Phi\}]] \tag{32}$$

The second equality follows from Schur's lemma (Thm A.1) assuring that $\Phi$ restricted to each $V_k$ is an isomorphism for each $k \in \mathcal{I}_\Phi$. We see in this expression that if $b \in \mathcal{I}_\Phi$, then $\min_\Psi R_b(\Psi, \Phi) = 0$ because $\mathbb{E}[g \circ V_b \mid \{g \circ V_k; k \in \mathcal{I}_\Phi\}] = g \circ V_b$ whenever $b \in \mathcal{I}_\Phi$. We therefore focus on $b \notin I_\Phi$ and show that $R_b$ with $b \notin I_\Phi$ scales with $\mathbb{E}[\|V_b\|^2]$. First, note that because the norm on $\mathcal{X}$ is based on $G$-invariant inner product, $\mathbb{E}[\|g \circ V_b\|^2] = \mathbb{E}[\|V_b\|^2]$. Next note that, for any scalar $a > 0$, we have

$$\mathbb{E}[g \circ aV_b \mid \{g \circ V_k; k \in \mathcal{I}_\Phi\}] = a\mathbb{E}[g \circ V_b \mid \{g \circ V_k; k \in \mathcal{I}_\Phi\}]. \tag{33}$$

Therefore

$$\min_\Psi \mathbb{E}_{X \sim \mathcal{P}, g \sim \mathcal{P}_G}[\|g \circ aV_b - P_b\Psi\Phi(g \circ X)\|^2] = \mathbb{E}[\|g \circ aV_b - \mathbb{E}[g \circ aV_b \mid \Phi(g \circ X)]\|^2] \tag{34}$$

$$= a^2 \mathbb{E}\|g \circ V_b - \mathbb{E}[\|g \circ V_b \mid \Phi(g \circ X)]\|^2] \tag{35}$$

$$= a^2 \min_\Psi R_b(\Phi, \Psi | \mathcal{P}, \mathcal{P}_G) \tag{36}$$

Thus there is a distribution $\mathcal{P}_1^b$ of $X$ with $\mathbb{E}_{\mathcal{P}_1^b}[\|V_b\|^2] = 1$ such that, by factoring out $\mathbb{E}_{\mathcal{P}}[\|V_b\|^2]$ as we did $a$ in the expression above, we can obtain

$$\min_\Psi R_b(\Phi, \Psi | \mathcal{P}, \mathcal{P}_G) = \mathbb{E}_{\mathcal{P}}[\|V_b\|^2]\left(\min_\Psi R_b(\Phi, \Psi | \mathcal{P}_1^b, \mathcal{P}_G)\right) \tag{37}$$

indicating that $\min_\Psi R_b(\Phi, \Psi | \mathcal{P}, \mathcal{P}_G)$ scales with $\mathbb{E}_\mathcal{P}[\|V_b\|^2]$ for every choice of $\mathcal{I}_\Phi$. Finally, see from eq.(32) that $\min_\Psi R_b(\Phi, \Psi | \mathcal{P}_1, \mathcal{P}_G)$ depends only on $\mathcal{I}_\Phi$, so define $\mathcal{R}_b(\mathcal{I}_\Phi) := \min_\Psi R_b(\Phi, \Psi | \mathcal{P}_1, \mathcal{P}_G)$. Because $\Psi$ can be chosen on each $V_b$ separately for each $b$, we see that

$$L(\Phi) = \sum_{b \notin \mathcal{I}_\Phi} \mathbb{E}_\mathcal{P}[\|V_b\|^2] \mathcal{R}_b(\mathcal{I}_\Phi). \tag{38}$$

$\square$

From this result we see that the optimal solution the optimal $\Phi^*, \Psi^*$ in the linear case can be found by minimizing $\sum_{b \notin \mathcal{I}_\Phi} \mathbb{E}_\mathcal{P}[\|V_b\|^2] R_b(\mathcal{I}_\Phi)$ about $\mathcal{I}_\Phi$ with $\Phi \in F$ so that $\sum_{b \in \mathcal{I}_\Phi} \dim(V_b) \leq \dim(W)$ and find the corresponding $\Psi$ by setting $\Psi_b(V) = P_b \Psi(V) = \mathbb{E}[V_b | \{V_k; k \in \mathcal{I}_\Phi\}]$. This is indeed an optimization problem about the discrete set $\mathcal{I}_\Phi$. We also claim the following about the dependency of the solution to this problem on the norms of $V_b$.

**Corollary C.9.** *For $J \subset \mathcal{I}_\Phi$, define $\ell(J) = \sum_{b \in J} \mathbb{E}_{X \sim \mathcal{P}} E[\|V_b\|^2]$. For any $\Phi_1$ and $\Phi_2$, we have $L(\Phi_1) > L(\Phi_2)$ if (1) $\mathcal{I}_{\Phi_1} \subset \mathcal{I}_{\Phi_2}$ or (2) $\ell(\mathcal{I}_{\Phi_1^c})$ is sufficiently larger than $\ell(\mathcal{I}_{\Phi_2^c})$.*

*Proof.* If $\mathcal{I}_{\Phi_1} \subset \mathcal{I}_{\Phi_2}$, then note that

$$\begin{aligned} L(\Phi) = \min_\Psi \mathcal{L}(\Phi, \Psi | \mathcal{P}, \mathcal{P}_G) &= \mathbb{E}[\|g \circ X - \Psi\Phi(g \circ X)\|^2] \\ &= \mathbb{E}[\|g \circ X - E[g \circ X | \Phi(g \circ X)]\|^2] \\ &= \mathbb{E}[g \circ V - \mathbb{E}[g \circ V \mid \{g \circ V_k; k \in \mathcal{I}_\Phi\}]], \end{aligned} \tag{39}$$

Because $\sigma(\{g \circ V_k; k \in \mathcal{I}_{\Phi_2}\}) \supset \sigma(\{g \circ V_k; k \in \mathcal{I}_{\Phi_1}\})$ where $\sigma$ is the sigma field, it trivially follows that $L(\Phi_1) > L(\Phi_2)$ by the definition of the conditional expectation.

Next, if $\mathcal{I}_{\Phi_1} \not\subset \mathcal{I}_{\Phi_2}$, note that $L(\Phi) = \min_{\Psi \in \mathcal{C}_\Phi(\mathcal{X})} \mathcal{L}(\Phi, \Psi | \mathcal{P}, \mathcal{P}_G)$ satisfies

$$c_u(\mathcal{I}_\Phi) \sum_{b \notin \mathcal{I}_\Phi} \mathbb{E}_\mathcal{P}[\|V_b\|^2] \geq L(\Phi) = \sum_{b \notin \mathcal{I}_\Phi} \mathbb{E}_\mathcal{P}[\|V_b\|^2] R_b(\mathcal{I}_\Phi) \geq c_l(\mathcal{I}_\Phi) \sum_{b \notin \mathcal{I}_\Phi} \mathbb{E}_\mathcal{P}[\|V_b\|^2] \tag{40}$$

for some $c_u$ and $c_l$. Thus, if

$$\sum_{b \notin \mathcal{I}_{\Phi_1}} \mathbb{E}_\mathcal{P}[\|V_b\|^2] > \frac{c_u(\mathcal{I}_{\Phi_2})}{c_l(\mathcal{I}_{\Phi_1})} \sum_{b \notin \mathcal{I}_{\Phi_2}} \mathbb{E}_\mathcal{P}[\|V_b\|^2] \tag{41}$$

Then

$$L(\Phi_1) \geq c_l(\mathcal{I}_{\Phi_1}) \sum_{b \notin \mathcal{I}_{\Phi_1}} \mathbb{E}_\mathcal{P}[\|V_b\|^2] \geq c_u(\mathcal{I}_{\Phi_2}) \sum_{b \notin \mathcal{I}_{\Phi_2}} \mathbb{E}_\mathcal{P}[\|V_b\|^2] > L(\Phi_2) \tag{42}$$

Then $L(\Phi_1) > L(\Phi_2)$ is guaranteed. In other words, We have $L(\Phi_1) > L(\Phi_2)$ when $\ell(\mathcal{I}_{\Phi_1^c})$ is sufficiently larger than $\ell(\mathcal{I}_{\Phi_2^c})$.

$\square$

### C.3.2 NONLINEAR FOURIER FILTER

In this section, we extend the argument in the previous section to a more general situation in which $\mathcal{X}$ is not necessarily a representation space itself. Before we proceed, a word of caution is in order regarding the space on which the encoder $\Phi$ is to be searched in the optimization process.

As is the case in many applications, we set our goal to find $\Phi$ from a set of functions with certain smoothness, and we use this smoothness property to relate the harmonic information captured in the latent space. In analogy to this argument, we thus restrict the search space of our encoder $\Phi$ to a space of functions with a certain level of smoothness in order to make the claims meaningful. Therefore, in this section, we assume that there exists a $G$-invariant kernel $K(\cdot, \cdot)$, and that we are searching the encoder from the space of equivariant functions from $\mathcal{X}$ to some vector space $V$ over $\mathbb{R}$ satisfying the following properties:

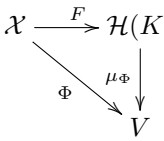

Figure 11: The commuting relation established by Lemma C.11

**Definition C.10.** Let $K$ be a $G$-invariant positive definite kernel on $\mathcal{X}$, and $\mathcal{H}(K)$ be the RKHS corresponding to $K$. A map $\Phi : \mathcal{X} \to V$ for some Hilbert space $V$ is called a $\mathcal{H}(K)$-*continuous equivariant map* if the following two conditions are satisfied;

1. The action of $g$ on $\Phi(x)$ is linear; that is, there exits a representation $D_\Phi : G \to GL_\mathbb{R}(V)$ such that $D_\Phi(g)\Phi(x) = \Phi(g \circ x)$.

2. For all $v \in V$, the map $x \mapsto \langle v, \phi(x) \rangle_V$ belongs to $\mathcal{H}(K)$.

We first show that any $\mathcal{H}(K)$-continuous equivariant map factors through the map from $\mathcal{X}$ to $\mathcal{H}(K)$ (Fig 11).

**Lemma C.11.** *Suppose that $K$ is a $G$-invariant positive definite kernel on $\mathcal{X}$ and $\Phi$ is a $\mathcal{H}(K)$-continuous equivariant map from $\mathcal{X}$ to $V$. For the canonical embedding $F : \mathcal{X} \to \mathcal{H}(K)$, $x \mapsto k_x$, there exists a linear equivariant map $\mu_\Phi : \mathcal{H}(K) \to V$ such that $\Phi = \mu_\Phi \circ F$, where $\mathcal{H}(K)$ admits the linear regular action $\rho(g) : f \mapsto f(g^{-1} \cdot)$.*

*Proof.* Because $\langle v, \Phi(\cdot) \rangle_V \in \mathcal{H}(K)$ for any $v$ by assumption, there is $f_v^\Phi \in \mathcal{H}(K)$ such that $\langle v, \Phi(x) \rangle = \langle f_v^\Phi, k_x \rangle_H$, where $\langle \, , \, \rangle_H$ is the inner product of $\mathcal{H}(K)$. This being said, if $\{e_i\}$ is the orthonormal basis of $V$, we have

$$\Phi(x) = \sum_i e_i \langle e_i, \Phi(x) \rangle_V$$

$$= \sum_i e_i \langle f_i^\Phi, k_x \rangle_H$$

Motivated by this expression, let us define the linear map $\mu_\Phi : \mathcal{H}(K) \to V$ by

$$\mu_\Phi : h \mapsto \sum_i e_i \langle f_i^\Phi, h \rangle_H. \tag{43}$$

We claim that the diagram Fig 11 commutes and $\mu_\Phi$ is equivariant.

First, it follows from eq.(43) that

$$\mu_\Phi(k_x) = \sum_i e_i \langle f_i^\Phi, k_x \rangle_H = \sum_i e_i \langle e_i, \Phi(x) \rangle_V = \Phi(x), \tag{44}$$

which means $\mu_\Phi \circ F(x) = \Phi(x)$. Recall that the linear action $\rho$ is defined by $\rho(g) : h(\cdot) \mapsto h(g^{-1} \cdot)$. Then, we have $\rho(g)k_x = K(x, g^{-1} \circ \cdot) = K(g \circ x, \cdot) = k_{g \circ x}$, and thus, using eq.(44),

$$\mu_\Phi(\rho(g)k_x) = \mu_\Phi(k_{g \circ x})$$
$$= \Phi(g \circ x)$$
$$= D(g)\Phi(x).$$

Because the span of $\{k_x \mid x \in \mathcal{X}\}$ is dense in $\mathcal{H}(K)$, the above equality means the equivariance of $\mu_\Phi$. $\square$

Now, with Lemma C.11, we may extend the definition of multiplicity free map $\Phi$ that is $\mathcal{H}(K)$-continuous equivariant map.

**Definition C.12.** (Multiplicity-unlimited mapping, General) We say that a $\mathcal{H}(K)$-continuous equivariant map $\Phi : \mathcal{X} \to V$ is multiplicity unlimited if $\mu_\Phi$ constructed in the way of C.11 is a multiplicity unlimited map. Let us denote by $\mathcal{I}_\Phi$ the set of all irreducible types present in $V$. Also, let us use $P_b$ to denote the projection of $F(\mathcal{X})$ onto the $b$th isotypic component $\mathcal{V}_b$ of $F(\mathcal{X})$, and let $V_b = P_b F(X)$.

$$\mathcal{X} \lhook\joinrel\longrightarrow \mathcal{H}(K) \cong F_\Phi \oplus F_\Phi^c \xrightarrow{\phantom{xx}\mu_\Phi\phantom{xx}} F_\Phi \cong V$$

With this definition in mind, we can extend the argument in Proposition C.11 to the nonlinear case if the $F$ part of the decomposition $\Phi = \mu_\Phi \circ F$ is a $c_F$-BiLipschitz injective map.

**Theorem C.13.** *Suppose that there exists a $G$-invariant kernel $K$ on $\mathcal{X}$ for which the map $F : x \to K(x, \cdot)$ is a $c_F$-biLipschitz injective map, and define $\mathcal{M}(K)$ to be the set of all $\mathcal{H}(K)$-continuous equivariant, multiplicity unlimited maps $\Phi$ into a vector space $V$ with a linear action. We use $F_b$ to denote the map of $\mathcal{X}$ into $b$-th isotypic component of $\mathcal{H}(K)$. For any given $\Phi \in \mathcal{M}(K)$, define $L(\Phi) = \min_{\Psi \in \mathcal{C}(\mathcal{V})} \mathcal{L}(\Phi, \Psi | \mathcal{P}, \mathcal{P}_G)$, where $\mathcal{C}(\mathcal{X})$ is the set of all measurable maps from $V$ to $\mathcal{X}$. Also, define $\ell(J) = \sum_{b \in J} \mathbb{E}_{X \sim \mathcal{P}} E[\|V_b\|^2]$ with $V_b$ being the $b$th isotypic component of $F(X)$ and assume that there is some $\delta$ such that $E[\|V_b\|^2] > \delta$. Then for any $\Phi_1$ and $\Phi_2$ with $\mathcal{I}_{\Phi_1} \neq \mathcal{I}_{\Phi_2}$ we have $L(\Phi_1) > L(\Phi_2)$ if (1) $\mathcal{I}_{\Phi_1} \subset \mathcal{I}_{\Phi_2}$ or (2) $\ell(\mathcal{I}_{\Phi_1}^c)$ is sufficiently larger than $\ell(\mathcal{I}_{\Phi_2}^c)$.*

*Proof.* Using the similar logic as in the linear case, we establish the claim by showing that $L(\Phi) := \min_\Psi \mathcal{L}(\Phi, \Psi | \mathcal{P}, \mathcal{P}_G)$ is bounded from above and below by

$$\sum c_u(\mathcal{I}(\Phi))\mathbb{E}[\|V_b\|^2] \geq L(\Phi) \geq \sum c_l(\mathcal{I}(\Phi))\mathbb{E}[\|V_b\|^2] \tag{45}$$

for some $c_l$ and $c_u$. Assume the same $F$ used in Lemma C.11, and let $F_b$ be the $b$th isotypic component of $F$. To show the lower bound, by BiLipschitz property of $F$ we have

$$\min_\Psi \mathcal{L}(\Phi, \Psi | \mathcal{P}) \geq c_F \min_\Psi \mathbb{E}[\|F(g \circ X) - F(\Psi\Phi(g \circ X))\|^2] \tag{46}$$

$$\geq c_F \min_\Psi \sum_b \mathbb{E}[\|F_b(g \circ X) - F_b(\Psi\Phi(g \circ X))\|^2] \tag{47}$$

$$\geq c_F \sum_b \min_\Psi \mathbb{E}[\|F_b(g \circ X) - F_b(\Psi\Phi(g \circ X))\|^2] \tag{48}$$

$$\geq c_F \sum_{b \notin \mathcal{I}_\Phi} \mathbb{E}[\|g \circ V_b - \mathbb{E}[g \circ V_b | V_b; b \in \mathcal{I}_\Phi]\|^2] \tag{49}$$

where in the last inequality, we used the optimality of conditional expectation and the fact that in the factorization $\Phi = \mu_\Phi F$, $\mu_\Phi(X)$ has the same sigma algebra as $\{V_b; b \in \mathcal{I}_\Phi\}$ under the multiplicity unlimited setting. Because the formula above is of the same form as eq.(39), we are done with the lower bound here with the same logic as in the linear case. As for the Upper bound, again we have from the BiLipschitz property that

$$L(\Phi) = \min_\Psi \mathcal{L}(\Phi, \Psi | \mathcal{P}) \leq c_F \min_\Psi \mathbb{E}[\|F(g \circ X) - F(\Psi\Phi(g \circ X))\|^2] \tag{50}$$

$$= c_F \min_\Psi \sum_b \mathbb{E}[\|F_b(g \circ X) - F_b(\Psi\Phi(g \circ X))\|^2] \tag{51}$$

Now, noting that $\Phi = \mu_\Phi F$ under multiplicity unlimited setting is injective on its image and writing $F_\Phi = P_\Phi F$ to be the projection of $F$ onto the representations corresponding to $\mathcal{I}_\Phi$, $\Psi$ can be chosen to an invertible map that maps the image of $\Phi$ back to its preimage so that, by using the fact that $\mu_\Phi$ is invertible on $\mathcal{I}_\Phi$ component, we have $F_\Phi(x) = F_\Phi\Psi\Phi(x)$ for $x \in \mathcal{X}$.

To construct such $\Psi$, write $\Phi = \mu_\Phi F$, and for some $x \in \mathcal{X}$, consider $\mu_\Phi^{-1}(\Phi(x))$, which is possibly a set. Here, we may take a specific section $A_\phi(x) \in F(\mathcal{X})$ of $\mu_\Phi$, and define $\Psi$ so that $\Psi(\Phi(x)) = F^{-1}A_\phi(x)$ because $F$ maps $\mathcal{X}$ to its image injectively. This way, $F\Psi\Phi(x) = A_\phi(x)$ and $F_\Phi\Psi\Phi(x) = P_\Phi A_\phi(x)$. Because the choice of the section for each $x$ is arbitrary, let us take the section such that the norm of its projection $P_\Phi^c A_\phi(x) := P_\Phi^c F\Psi\Phi(x)$ is minimal amongst all members of $\mu_\Phi^{-1}(\Phi(x))$, where $P_\Phi^c$ is the projection to $\{\mathcal{V}_b | \mathcal{I}_\Phi\}$. This way, we have $\|F_b\Psi\Phi(x)\| \leq \|F_b(x)\|$ whenever $b \notin \mathcal{I}_\Phi$.

As for when $b \in \mathcal{I}_\Phi$, note by the definition $\mu_\Phi A_\phi(x) = \Phi(x) = \mu_\Phi F(x) = \mu_\Phi F_\Phi(x)$ and by Schur's lemma, $\mu_\Phi A_\phi(x) = \mu_\Phi P_\Phi A_\phi(x)$, so $\mu_\Phi P_\Phi A_\phi(x) = \mu_\Phi F_\Phi(x)$. Because $\mu_\Phi$ is invertible on $F_\Phi(\mathcal{X})$ by Schur's lemma and multiplicity-free condition, this shows that $F_\Phi(x) = P_\Phi A_\phi(x)$ because $P_\Phi A_\phi(x) \in F_\Phi(\mathcal{X})$. Altogether we have $F_\Phi\Psi\Phi(x) = F_\Phi(x)$ as desired.

With such $\Psi$, we have $F_b\Psi(\Phi(x)) = F_b(x)$ whenever $b \in \mathcal{I}_\Phi$ as well. Thus, for such a chosen $\Psi$, $L(\Phi)$ may be bounded from above by

$$c_F \sum_{b \notin \mathcal{I}_\Phi} \mathbb{E}[\|g \circ V_b - F_b(\Psi\Phi(g \circ X))\|^2] \tag{52}$$

Now, note that

$$\sum_{b \notin \mathcal{I}_\Phi} \mathbb{E}[\|g \circ V_b - F_b(\Psi\Phi(g \circ X))\|^2] \leq \sum_{b \notin \mathcal{I}_\Phi} \mathbb{E}[\|g \circ V_b\|^2] + \mathbb{E}[\|F_b(\Psi\Phi(g \circ X))\|^2] \tag{53}$$

$$= 2 \sum_{b \notin \mathcal{I}_\Phi} \mathbb{E}[\|g \circ V_b\|^2] \tag{54}$$

$$= 2 \sum_{b \notin \mathcal{I}_\Phi} \mathbb{E}[\|V_b\|^2] \tag{55}$$

by the choice of the section we chose to construct $\Psi$. This would define the upper bound $\sum_{b \notin \mathcal{I}_\Phi} 2c_F\mathbb{E}[\|V_b\|^2]$ and the same logic as in the linear case follows from here. $\qquad\square$

This statement tells us that, if a certain set of isotypes captures significantly more dominant information than others, then NFT will prefer the very set of isotypes over others. Indeed, if $\ell(\mathcal{I}^c_{\Phi_1})$ is sufficiently larger than $\ell(\mathcal{I}^c_{\Phi_2})$, it will mean that the set of isotypes that $\Phi_1$ is excluding from $H(K)$ is *so* much larger in the dataset than those excluded by $\Phi_2$ that it is more preferable to choose $\Phi_2$ than $\Phi_1$. This way, the solution to the autoencoding problem is determined by the contribution of each frequency space to the dataset.

# D  DETAILS OF EXPERIMENTS

## D.1  COMPRESSION OF DEFORMED DATA

In this section we provide the details of the signal compression experiment in Sec 5.1, along with the additional visualization of the reconstructed results.

**Dataset:** The dataset was constructed as described in Sec 5.1, and the velocity of the shift was chosen from the range $[0, 64]$.

**Models:** For the encoder and decoder, MLP with two hidden layers is used. The intermediate dimensions and the activations of the network are as follows:
$g$-NFT ($g$ known): Encoder 128-256-256-32, Decoder 32-256-256-128. activation=ReLU.
$G$-NFT ($G$ known, but $g$ unknown): Encoder 128-512-512-32, Decoder 32-512-512-128. activation=Tanh.

**Visualization of the reconstructed signals:**

(A) Blue: Reconstruction with NFT ($g$ known). Red: noisy training data.

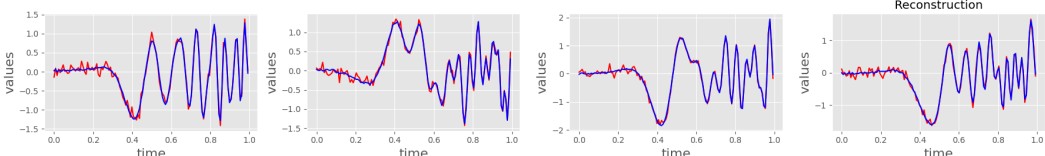

(B) Blue: Reconstruction with DFT. Red: Noiseless test data.

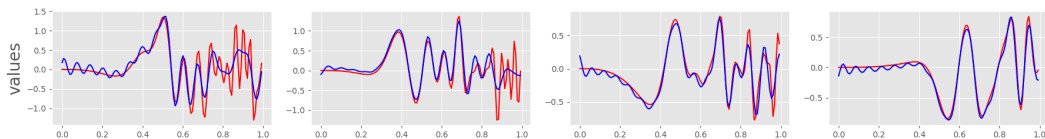

Figure 12: Reconsturction of NFT and DFT for nonlinearly transformed functions. Times in these figures are scaled by $1/128$.

As we can see in Fig 12, by the nonlinear transform $t \mapsto t^3$, the left area around the origin is flattened, and the right area around $t = 1$ is compressed. As a result, the deformed signal has a broader frequency distribution than the original latent signal (see Fig 3 also). The results in Fig 12 depict that the standard DFT, which is applied directly to the deformed signal, fits the left area with undesirable higher frequencies, while fitting the right area with an over smoothed signal. The NFT accurately reconstructs the signal in all the areas.

## D.2  1D SIGNAL HARMONIC ANALYSIS SEC 5.1

In this section we provide more details of the 1d signal experiment, along with the more detailed form of the objective we used to train the encoder and decoder. We also provide more visualizations of the spectral analysis (Fig 14).

**Dataset:** The dataset was constructed as described in Sec 5.1, and the velocity of the shift was chosen from the range $[0, 64]$, which suffices for the computation of the character inner product for real valued sequence of length 128.

**Model**: For both encoder and decoder, we used the same architecture as in D.1. However, the latent space dimension was set to be $10 \times 16$, which is capable of expressing at most $10/2 = 5$ frequencies.

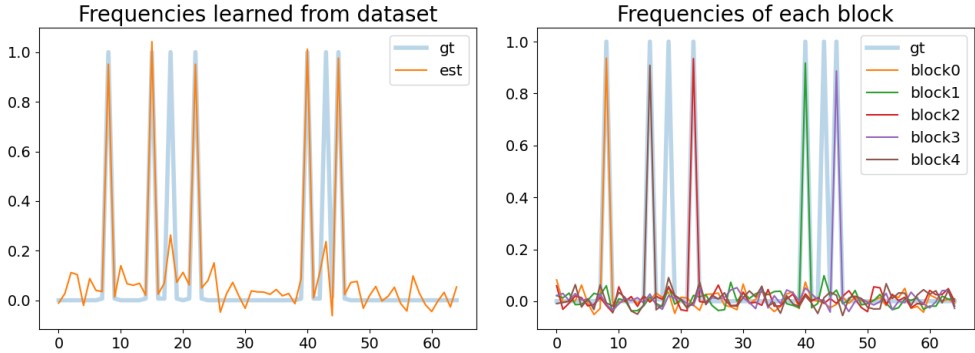

Figure 13: The plot of $\mathbb{E}[\langle M|\rho_f\rangle]$ (Left) and the plot of $\mathbb{E}[\langle B_i|\rho_f\rangle]$ for each block of the block-diagonalized $Ms$(right) when the major frequencies are $\{8, 15, 22, 40, 45\}$ and minor frequencies with weak coefficients are $\{18, 13\}$. We can confirm in the plot that major spikes only occur at major frequencies, and that each block corresponds to a single major frequency. Note that, when there are noise frequencies, they are slightly picked up in the overall spectrum of $M$ (Left). However, as seen in Sec 5.1, each block in the block diagonalization of $M$ contributes only slightly to the noise frequencies, distributing the noise over the entire $M$ instead of corrupting a specific dominant frequency. With this character analysis, we can identify the major frequencies almost perfectly by using an appropriate threshold (Fig 14).

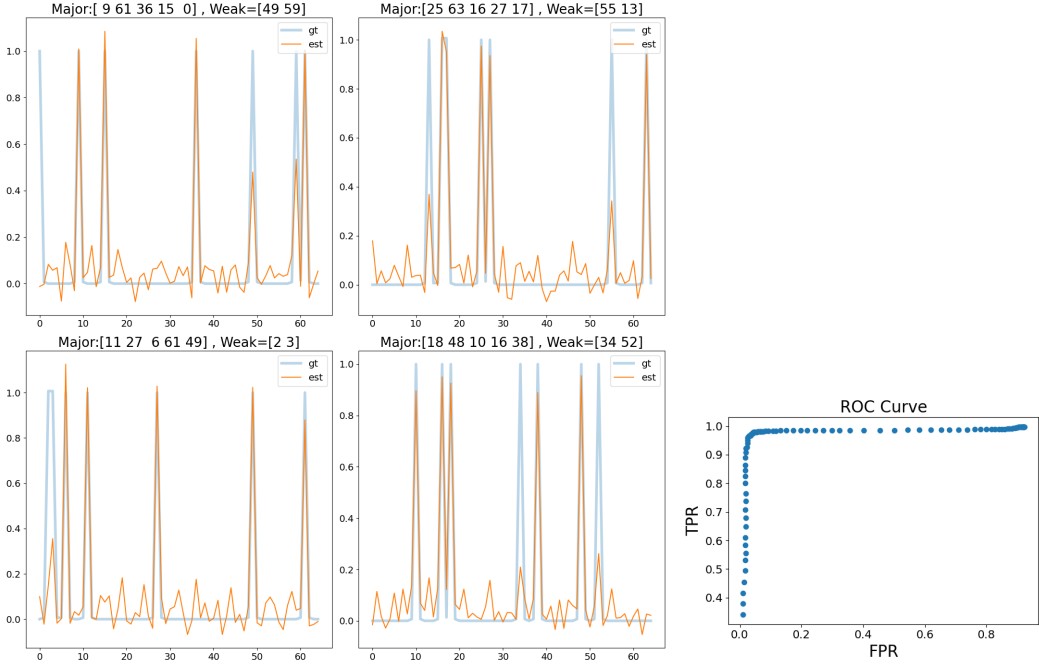

Figure 14: (Left)Other visualizations of $\mathbb{E}[\langle M|\rho_f\rangle]$ plotted against $f$ on the deformed signal datasets generated with different sets of major/minor frequencies. On any one of these plots, we can identify the major frequencies by simply looking for the set of $f$s for which $\mathbb{E}[\langle M|\rho_f\rangle]$ is above some threshold. (Right) ROC curve of the major-frequency identification with different thresholding values, computed over 100 datasets with randomly selected 5 major frequencies and 2 minor frequencies. Note that, on this setting of frequency detection, there are always 5/64 frequencies that are "positive". When normalized by (1-5/64), AUC was 0.97.

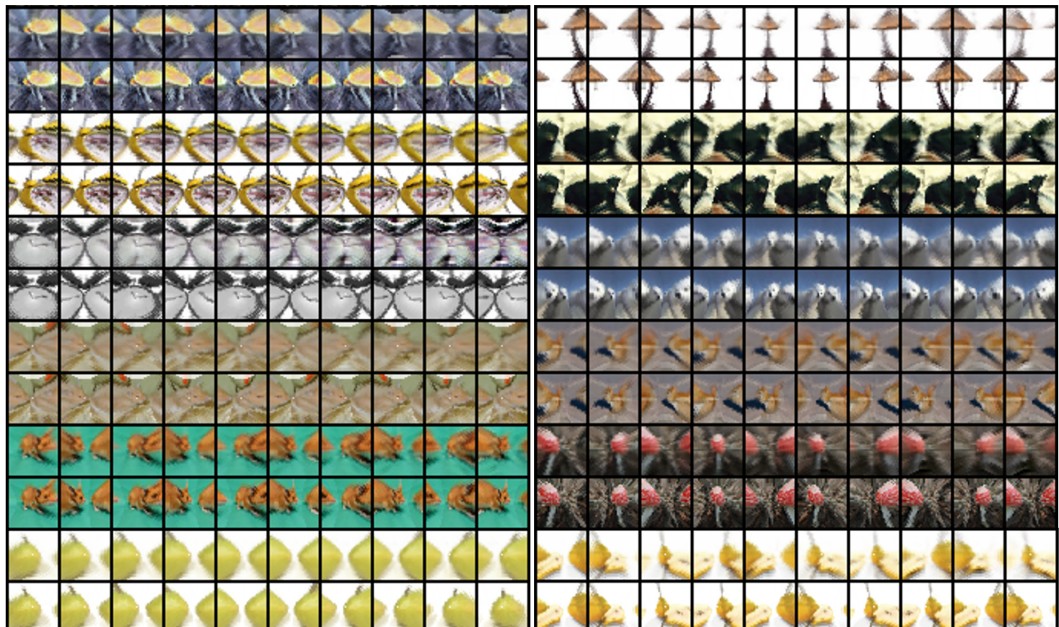

Figure 15: Prediction of shift under fish-eye deformation. Each predicted sequence is placed above the ground truth sequence. The predictions are computed as $\Psi(M^k\Phi(x_0))$, where $M$ is being regressed from the encoding of the first two frames $(x_0, x_1)$.

### D.3 PREDICTION OF THE SHIFT UNDER FISHEYE DEFORMATION IN SEC 5.1

We provide the details of the experiment on fisheye-deformed shift in Sec 5.1. We also provide more visualizations of the predicted sequence (Fig 15).

**Dataset**: For the fisheye prediction, the sequence of images was constructed from Cifar100 by first applying a sequence of horizontal shifts with random velocity in the range of $0 : 16$ pixels, and by applying the fisheye transformation (defisheye Contributors, 2019).

**Model**: For the encoder and decoder, ViT architecture was used with mlp-ratio=4, depth=8, number of heads=6 and embed dimension = 256, and the model was trained over 200000 iterations with Adam (lr=0.01). We decayed the learning rate linearly from 80000-th iteration. In the training of encoder and decoder in U-NFT, the transition matrix $M$ from the first to the second frame was validated at the third and the fourth frame to compute the loss (Same as in eq.(5) with $T = 2$, but used $\mathbb{E}[\|x_2 - \Psi(M_*(x_1))\|^2 + \|x_3 - \Psi(M_*^2(x_1))\|^2]$ to optimize $(\Phi, \Psi)$).

### D.4 DETAILS OF SO(2) EXPERIMENTS (SECTION 5.2)

In this section we provide the details of the rotated MNIST experiment in Section 5.2. In this set of experiments, we used OOD task to compare the representation learned by the encoder $\Phi$ against other representation learning method as well as supervised methods. As representation learning methods, we trained an autoencoder (AE), SimCLR (Chen et al., 2020), the invariant autoencoder (IAE) (Winter et al., 2022) and MSP (Miyato et al., 2022). We used two-layer CNN for both the encoder (and decoder) of these models. To evaluate these methods, we first trained the encoder $\Phi$ on the in-distribution training set, and then trained the linear classifier on the out of distribution dataset by optimizing

$$CrossEntropy(Softmax(W^T\Phi(X)), Y)$$

with respect to $W$ using **the fixed** $\Phi$, where $Y$ is the label of $X$. In Figure 7, we reported the size of the out-of-distribution dataset used to train $W$ as *num-finetune-data*.

For the supervised methods to compare against NFT, we used the two-layer CNN, $C_n$ (cyclic group) steerable CNN (Cohen and Welling, 2017), and $SO(2)$ steerable CNN Weiler et al. (2018). For the training of these models, we simply trained the cross entropy loss on the in-distribution training set,

and fine-tuned the final softmax layer on the out-of-distribution dataset to evaluate the OOD performance. In particular, we simultanesouly trained **both** $W$ and the parameters of *Net* to minimize

$$CrossEntropy(Softmax(W^T Net(X)), Y)$$

on the in-distribution set, and exclusively fine-tuned $W$ on the out-of-distribution set.

For steerable networks, we implemented them based on escnn library (Cesa et al., 2022) We used the same architecture described in
`https://github.com/QUVA-Lab/escnn/blob/master/examples/model.ipynb`
for $C_n$ CNN and `https://uvadlc-notebooks.readthedocs.io/en/latest/`
`tutorial_notebooks/DL2/Geometric_deep_learning/tutorial2_`
`steerable_cnns.html#SO(2)-equivariant-architecture` for $SO(2)$ CNN.

For g-NFT and MSP, we reshaped the $d_a d_m$-dimensional output of the encoder into a $d_a \times d_m$ matrix so that it is compatible with the group representation matrix $M \in \mathbb{R}^{d_a \times d_a}$ acting from left. For g-NFT, because we know the irreducible representations of SO(2), we modeled the transition matrix in the latent space as representation $M(\theta)$ with frequency up to $l_{\max} = 2$. We thus used the same parametrization as in the compression experiment of Sec 5.1 except that we used $1 \times 1$ identity matrix for the frequency 0, producing $(2l_{\max} + 1) \times (2l_{\max} + 1)$ matrix. We then trained $(\Phi, \Psi)$ with $\Phi$ having the latent dimension $\mathbb{R}^{(2l_{\max}+1) \times d_m}$. We trained each model for 200 epochs, which took less than 1 hour with a single V100 GPU. We used AdamW optimizer (Loshchilov and Hutter, 2017) with $\beta_1 = 0.9$ and $\beta_2 = 0.999$.

The hyperparameter space and the selected hyperparameters for each method were as follows:

- **autoencoder** learning rate (LR): 1.7393071138601898e-06, weight decay (WD): 2.3350294269455936e-08
- **supervised** LR: 0.0028334155436987537, WD: 4.881328098770217e-07
- **SimCLR** (Chen et al., 2020) projection head dim: 176, temperature: 0.3296851654086481, LR: 0.0005749009519935808, WD: 2.7857915542790035e-08
- **IAE** (Winter et al., 2022) LR: 0.0013771092749156428, WD: 1.2144233629665408e-06
- **Steerable(C_N)** angular resolution $n$: 28, LR: 0.002736272679599058, WD: 6.569644997770058e-06
- **Steerable(SO2)** maximum frequency: 4, LR: 0.003013815048663727, WD: 7.33786837736026e-06
- **MSP** (Miyato et al., 2022) $d_a$: 9, latent dimension $d_m$: 252, LR: 1.2124794217519062e-05, WD: 0.016388489985789633
- **g-NFT** maximum frequency $l_{\max}$: 2, $d_m$: 180, LR: 0.000543795556795646, WD: 0.00064916272403101113

The hyperparameters of each baseline, such as the learning rate and $l_{\max}$, were selected by Optuna (Akiba et al., 2019) based on the test prediction error on MNIST.

Although IAE is designed to be able to estimate each group action $g$, we supervised $g$ in this experiment to make a fair comparison to other $g$-supervised baselines. As the loss function for g-NFT, we used the sum of the autoencoding loss defined in Theorem 4.2 together with the *alignment loss*: $\|\Phi(g \circ X) - M(g)\Phi(X)\|^2$. This loss function promotes the equivariance of the encoder $\Phi$. See Fig 16 for more visualizations of the reconstruction with different values of $l_{\max}$.

### D.5  DETAILS OF 3D EXPERIMENTS (SECTION 5.2)

In this section we provide the details of the experiment on the rendering of 3D objects (ModelNet10-SO3, Complex BRDFs, ABO-Material). All the experiments herein were conducted with 4 V100 GPUs.

**ModelNet10-SO3**  The dataset in this experiment contains the object-centered images of 4,899 CAD models[3]. Each image was rendered with the uniformly random camera position $g \in SO(3)$.

---

[3]`https://github.com/leoshine/Spherical_Regression/tree/master/S3.3D_`
`Rotation`

The CAD models were separated into a training set (100 images for each CAD model) and a test set (20 images for each). We downscaled the image resolution to $64 \times 64$. We used the vision Transformer (ViT) for the encoder and the decoder. The encoder consisted of 9 blocks of 512-dimensional embedding and 12 heads. The decoder consisted of 3 blocks of 256-dimensional embedding and 6 heads. The patch size was $4 \times 4$ and MLP-ratio was 4 for both networks. We set $d_m = 64$ and $l_{\max} = 8$. The encoder output was converted to the $d_a \times d_m$ matrix by EncoderAdapter module defined by the following PyTorch pseudocode.

```
from torch import nn
from einops.layers.torch import Rearrange

class EncoderAdapter(nn.Module):
    def __init__(self, num_patches, embed_dim, d_a, d_m):
        self.net = nn.Sequential(
            nn.Linear(embed_dim, embed_dim // 4),
            Rearrange('b n c -> b c n'),
            nn.Linear(num_patches, num_patches // 4),
            nn.GELU(),
            nn.LayerNorm([embed_dim // 4, num_patches // 4]),
            Rearrange('b c n -> b (c n)'),
            nn.Linear(embed_dim * num_patches // 16, d_a * d_m),
            Rearrange('b (m a) -> b m a', m=d_m),
        )

    def forward(self, encoder_output):
        return self.net(encoder_output)
```

We also used a similar network before the decoder to adjust the latent variable. We used AdamW optimizer with batch size $48$, learning rate $10^{-4}$, and weight decay $0.05$. We didn't use the alignment loss described in Sec D.4. We trained the model for 200 epochs, which took 3 days.

**Complex BRDFs** : The dataset in this experiment contains the renderings of ShapeNet objects from evenly placed 24 views[4]. The camera positions in this experiment are constrained on a circle with a fixed radius so the group action of moving the camera position can be interpreted as the action of $SO(2)$ rather than $SO(3)$. We used 80% of the objects for training and 20% for testing. Following the terminology of the ViT family[5], We used ViT-B/16 for the encoder and ViT-S/16 for the decoder. The learning rate was 3e-4. We trained the model for 100 epochs, which took 1.5 days. Other settings were the same as the ones we used in the ModelNet experiment.

**ABO-Material** : The dataset in this experiment contains 2.1 million rendered images of 147,702 product items such as chairs and sofas[6]. The images were rendered from 91 fixed camera positions along the upper icosphere. We reduced the original $512 \times 512$ resolution to $224 \times 224$. The dataset was randomly partitioned into training (80%), validation (10%), and test (10%). The encoder and decoder architectures were the same as for the BRDFs experiment. We trained the model for 400 epochs with batch size 36, which took 12 days.

# E   SUPPLEMENTARY EXPERIMENTS FOR ROTATED MNIST

To further investigate the capability of NFT, we also extended our experiments of D.4 to consider the cases of image occlusion as well. Specifically, we zeroed out three quadrants of the input images, so that only one quarter of the image is visible to the models at all time. This is a more challenging task because the actions are not invertible at pixel level. We trained the encoder and decoder in the NFT framework on this dataset, and conducted the same experiment as in D.4. As

---

[4]https://github.com/google-research/kubric/tree/main/challenges/complex_brdf
[5]https://github.com/google-research/vision_transformer
[6]https://amazon-berkeley-objects.s3.amazonaws.com

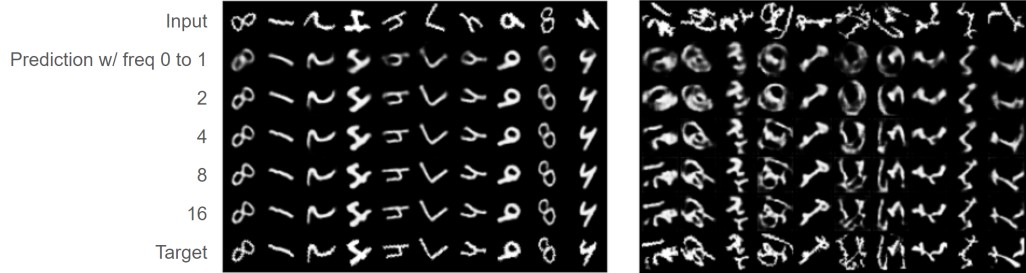

Figure 16: Reconstruction conducted with different values of $l_{max}$ ($1 \sim 16$). Higher frequencies promote sharper reconstruction of the images.

illustrated in Fig 17, the NFT-trained feature consistently demonstrated competitive performance (g-NFT, MSP).

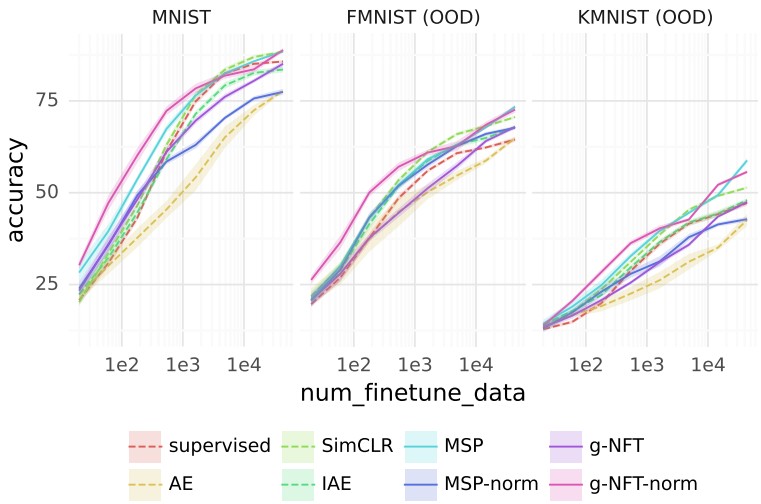

Figure 17: Linear probe results on Rotated MNIST with image occlusions.

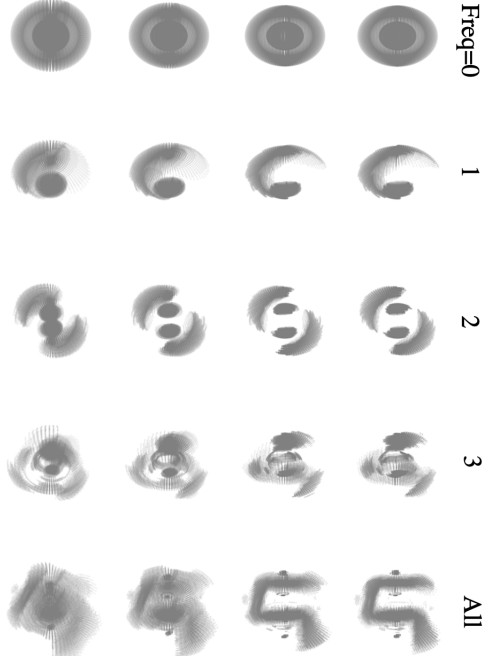

Figure 18: Spherical Decomposition of 3D point cloud representations of the same chair object in the leftmost panel of Fig 8, conducted in the way of (Skibbe et al., 2009). Although our experiment on the ModelNet dataset does not access the 3D voxel information nor include such structure in the latent space, each frequency in Fig 8 has much resemblance to the result of the point-cloud derived harmonic decomposition. See the movie folder in the Supplementary material for the movie rendition of this visualization (modelnet_chair_spherical.gif).

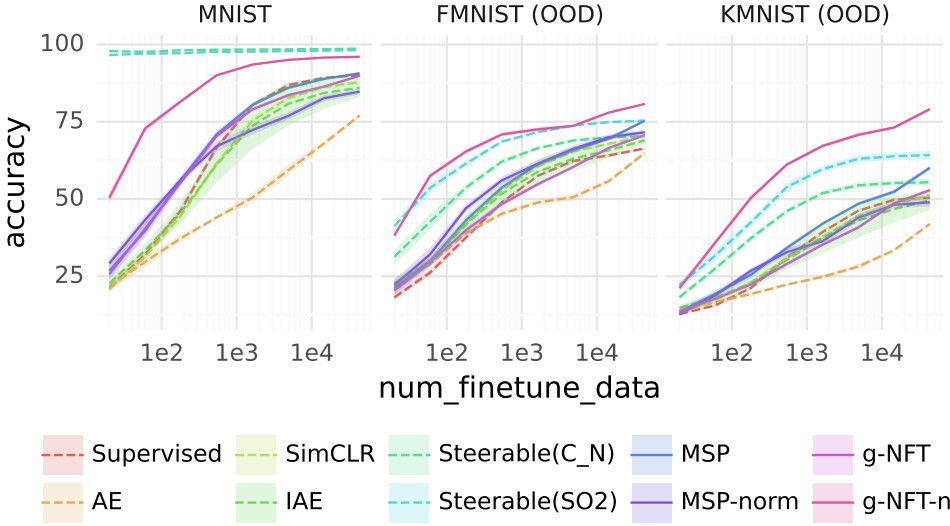

Figure 19: The result of the rotMNIST experiment (Fig.7) including the MSP and MSP-norm.

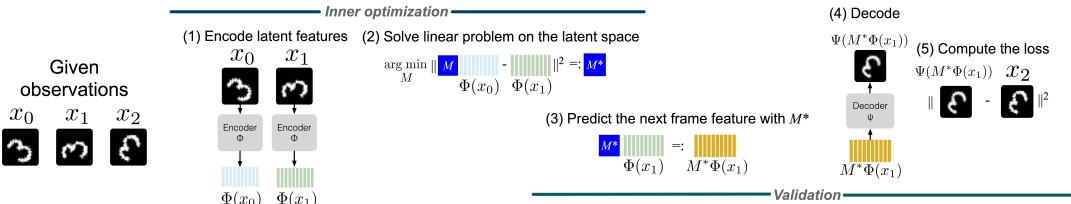

Figure 20: The meta sequential prediction (MSP) method used for U-NFT.

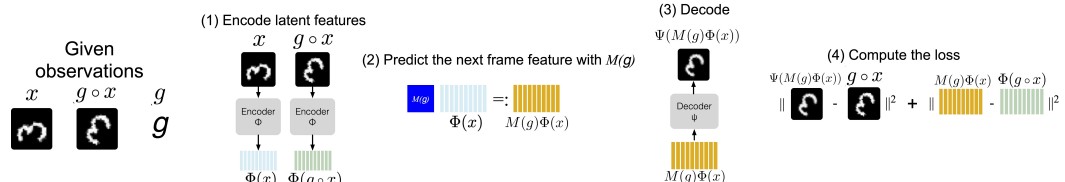

Figure 21: The training algorithm for g-NFT.

