# OpenReview forum: "Neural Fourier Transform: A General Approach to Equivariant Representation Learning"
_ICLR.cc/2024/Conference — ICLR 2024 poster_

### Official Review · Reviewer_ef5j · 2023-10-28

**Soundness:** 2 fair
**Presentation:** 3 good
**Contribution:** 2 fair
**Rating:** 6
**Confidence:** 3

**Summary:**

This paper proposes Neural Fourier Transform as a general approach for equivariant representation learning. The goal is to learn a latent linear action of the group, similar to how the DFT encodes linear representations of the shift group. Many symmetries (e.g. fisheye transformations) are obvious symmetries to the human eye, but there is no easy way to represent them using classic theory. The learning task is decomposed into three cases, where the network has to match using a linear action in latent space and a nonlinear action in data space. Theoretical results are verified on several tasks.

**Strengths:**

- The paper tackles the interesting task of representing nonlinear image transformations in a neural network as linear group actions. This is an interesting and valuable direction.
- The paper finds a simple yet effective strategy for solving the problem using tuples of $g$-transitions.
- While restricted to synthetic settings, the experiments are promising.

**Weaknesses:**

- The experiments could have been detailed a bit more clearly. I.e., what are the objectives, the loss functions used, the in/outputs to the model, etc.
- I find the title "Neural Fourier Transform" a bit misleading, as the Fourier transform is an extremely general signal processing tool, where in this work it is used as an example of finding linear representations of a group.
- It would have been nice to have some results on a task that is a bit more "in the wild".

**Questions:**

- How did you combine the autoencoder framework with the downstream (e.g. classification) objective? Did you have to weight the two objectives? How did you find the hyperparameters?
- Do you have an idea of how to get the tuples of nonlinear group actions in the wild?

---

> ### Author Response · Authors · 2023-11-17
> **Thank you very much for the comments and feedbacks!**
>
> Thank you very much for the feedback. We would like to respond to the weakness concerns and questions below.
>
> >The experiments could have been detailed a bit more clearly. I.e., what are the objectives, the loss functions used, the in/outputsmathematicall to the model, etc.
> How did you combine the autoencoder framework with the downstream (e.g. classification) objective? Did you have to weight the two objectives? How did you find the hyperparameters?
>
>
> We are sorry for the lack of clarity about the objectives in the experimental section. We changed the wording in the experimental section and  added more elaborations in the Appendix D.  As we describe in Appendix D, the inputs to $\Phi$ are time series in 5.1, and the inputs are image arrays in  5.2.
>
> In the OOD experiment of the classification of rotated images section 5.2, $(\Phi, \Psi)$ and classifiers are 'not' trained at the same time.  Instead, NFT is used as an unsupervised representation learning method, and we separately trained the linear classifier to be applied to the learned representation $\Phi(X)$.
> More particularly, we first trained the autoencoder $\Phi, \Psi$ using g-NFT on the pairs of images on the in-distribution dataset as described in section 3.  After then, with the 'fixed' trained pair $(\Phi, \Psi)$, we computed the logistic regression problem of minimizing the Cross Entropy Loss about the regression weight $W$ on the test (out of distributions) set.
> $$CrossEntropy(SoftMax(W^T\Phi(X)), Class(X)).$$
>
> In our comparison to SteerableCNN, however, we first trained the Steerable network by minimizing
> $$CrossEntropy(SoftMax(W^T SteerableNet(X)), Class(X))$$
> over both the parameters of the SteerableNet and $W$  on the training set, and “fine tuned” the $W$ part on the test set to evaluate the OOD performance.
>
> As we explain in the Appendix Section(D.4), we also optimized the hyperparameters using Optuna Software. We made the reference to Appendix in the main manuscript as well.
>
>
> > Do you have an idea of how to get the tuples of nonlinear group actions in the wild?
>
> We believe that tuples of nonlinear group actions can be obtained for example from the video sequences or, from the observation of dynamical systems in which the underlying group actions are group driven, such as robot arms or other physical systems in the wild.
> When the videos are sampled with relatively high temporal resolution, the transitions over triplet shall be small enough to be able to satisfy our constant velocity assumption over triplet/tuple $(x , g\circ x, g^2 \circ x)$
> Also, when the camera position is known in the 2D rendering of the 3D object, or more generally,  in the sim2real setting, we shall be able to obtain the information about the acting group element $g$ and hence obtain the dataset of form $(x, g\circ x)$ together with $g$ to perform g-NFT.

---

> ### Author Response · Authors · 2023-11-17
> **Thank you very much for the comments and feedbacks!  (2)**
>
> We would also like to try alleviating the following concern about our title.
>
> >I find the title "Neural Fourier Transform" a bit misleading, as the Fourier transform is an extremely general signal processing tool, where in this work it is used as an example of finding linear representations of a group.
>
> As we describe in Section 2,  we introduce NFT as an extension of FT, which is  mathematically formulated with linear representation of a group.
> In an effort to better justify NFT as a general extension of FT, we would like to further elaborate Section 2 by associating each component of DFT with a notion of group representation, and show that DFT is in fact about linear representations of a group.  First, we would like to recall that DFT is defined in signal processing as the linear map $\Phi : f \in C^T \to \hat{f} \in C^T$, where
>
> $f[j] =  \frac{1}{\sqrt{T}} \sum_{k=0}^{T-1} e^{2\pi i \frac{kj}{T}} \hat{f}[k] $ and
> $\hat{f}[k] =\frac{1}{\sqrt{T}} \sum_{j=0}^{T-1} e^{- 2\pi i \frac{kj}{T}} f[j] $
>
> and the map $\Phi$ is usually described by a DFT matrix $[DFT]$, so that $\Phi(f) = [DFT]f = \hat{f}$.
>
> ### Frequency:
> First, we would like to begin by interpreting DFT as a case of a change of basis operation from standard basis $B$ to a new basis $\hat B$.
> If the standard basis consists of $e_i$(one hot vector for $i$th coordinate), then $\hat B$ can be read from the entries of IDFT. Namely, the $k$th vector of  $\hat B$ is given by $v_k = \frac{1}{\sqrt{T}}[e^{ 2 \pi i kj /T} ; j=0:T]$.   Because the $k$th coordinate under DFT is called the $k$th frequency,  we are safe to identify $span(v_k)$ or $v_k$ itself with the $k$th frequency.
>
>
> As we explain above eq.(2) in the manuscript, $v_k$ has a special property with respect to “shift” action (cyclic group) . Namely, if $(m \circ v)[j] = v[ (j-m)mod\ T]$ is the shift by $m$ coordinates, then $m \circ v_k = e^{2\pi i m k /T} v_k$ for all $m$.
> Note that $m$-shift can be expressed as a shift matrix $[M_m]$ (permutation matrix) in the standard basis, and $m \to [M_m]$ is a group representation.  This means that $[M_m] v_k = e^{2\pi i m k /T} v_k$ and $v_k$ is an eigenvector for the shift operation of all sizes.  Thus the $k$th frequency can be regarded as a one dimensional $k$th eigenspace, which in group representation is called “irreducible representation”.
> This way, the $k$th frequency is the $k$th irreducible representation, and DFT emerges as decomposition of a signal into irreducible representations (frequencies) via
> $f  = \sum_k \hat f[k] v_k$.
>
> ### Filter:
> In signal processing, filter is used as a major tool in application of DFT, and its basic form functions by deciding a set of frequencies $S$ and using
> $f_S =\sum_{k \in S} \hat f[k] v_k $ to approximate $f$. When $S$ is chosen from low frequencies, this operation is called “low-pass filter”.  Now, with the association of the frequency to irreducible representation, this is equivalent to choosing the set of irreducible representations $S$.  In the context of classic FT, theorem 4.2 claims that when the output dimension of $\Phi$ is restricted and when $S$ is not specified, the MSE loss in NFT tends to train $(\Phi, \Psi)$ so that $\Psi \Phi (f) = f_S$, where $S$ is chosen to be able to best reconstruct the signal $f$ from $f_S$.
>
> ### Neural FT vs FT :
> So far, we have described FT as a transform that decomposes a signal into irreducible representations, when the action on the signal is linear/known. Now, as we state in our work,  the mission of our research is to decompose a signal into irreducible representations when the action is possibly nonlinear/unknown/partially known.
> When the action is nonlinear/unknown,  we need two modifications to the classic FT: (1) allowing $\Phi$ to be nonlinear and (2) learning the Fourier basis $\hat  B$ from the observations.  Indeed, when the action is known and linear, $\Phi$ in our framework agrees by definition with that of FT (More particularly, the case of g-NFT with linear action).   This is indeed the way we constructed our NFT.
>
> This discussion extends naturally to the case of continuous FT. Please also see (Clausen and Baum, Fast Fourier transforms, 1993) and   (Fourier Analysis on Groups, Walter Rudin) for more detailed  group-theoretic explanation of FT.
> We hope this explanation better justifies our title.

---

> > ### Comment · Reviewer_ef5j · 2023-11-22
> >
> > Thanks for your detailed feedback. I understand the reasoning behind the title, but I still believe the FT is a specific instance of methods that decompose signals into irreps. The paper presents a method that can be applied more generally and actually could have benefited from a more general title. Nevertheless, the authors justify their reasoning well. My score remains.

---

### Official Review · Reviewer_g5qF · 2023-10-31

**Soundness:** 3 good
**Presentation:** 3 good
**Contribution:** 3 good
**Rating:** 8
**Confidence:** 3

**Summary:**

This manuscript presents the Neural Fourier Transform (NFT), a novel framework for learning latent linear group actions without requiring explicit knowledge of how the group (or even what group) acts on the input data. The paper rigorously explores the theoretical underpinnings related to the existence of linear equivariant features. Moreover, empirical results are provided to substantiate the utility of NFT in a range of applications, each distinguished by differing levels of prior knowledge about the input group action.

**Strengths:**

1. The paper is well written and well structured. Many remarks and explanation are provided, making it easily comprehensible
2. The core concept—learning a latent group representation space by utilizing triples of group-acted inputs, without the need for prior information on the group—is both intriguing and innovative.
3. The empirical findings presented are both compelling and substantiate the framework's utility.

**Weaknesses:**

1. I have only one question/comment: in Theorem 4.2, $(\Phi^*, \Psi^*)$ is the minimizer of
$$E_{g\in G}[\|g\circ X - \Psi \Phi (g\circ X)\|^2]$$
Does this merely imply that $\Psi \circ\Phi $ is a "good" autoencoder on the group orbit, instead of transforming $X$ into a linear space on which there is a linear $G$ action? Should it be the following instead?
$$E_{g\in G}[\|g\circ X - \Psi \circ M(g) \circ \Phi ( X)\|^2]$$

**Questions:**

Please refer to the previous comment.

---

> ### Author Response · Authors · 2023-11-16
> **Thank you very much for the comments and feedbacks**
>
> Thank you very much for the comment! In Theorem 4.2 of the original manuscript, we stated to seek $(\Phi^*, \Psi^*)$ “among the set of autoencoders of a fixed latent dimension that can linearlize the actions in the latent space.”  We are sorry for the confusion that might have been caused by this expression. By the phrase “an autoencoder that can linearlize the actions”, we meant an equivariant autoencoder $(\Phi, \Psi)$ for which  there is $M(g)$ such that $\Phi(g \circ X) = M(g) \Phi(X)$ for each $g$.  Indeed, we are therefore discussing here the minimization of the loss over the restricted pool of equivariant $(\Phi, \Psi)$ with fixed latent dimensions and linear latent action. We reworded this part in the revision.

---

> > ### Comment · Reviewer_g5qF · 2023-11-21
> >
> > I thank the authors for the response. My rating remains.

---

### Official Review · Reviewer_XpFk · 2023-11-01

**Soundness:** 3 good
**Presentation:** 3 good
**Contribution:** 3 good
**Rating:** 8
**Confidence:** 3

**Summary:**

This paper proposes Neural Fourier Transform for equivariant representation learning that deals with the cases in which the group action on the data may be nonlinear or implicit. The paper investigates three scenarios based on the level of prior knowledge on the group and group element that acts on each data. Experimental results are provided to demonstrate the effectiveness of the proposed method in nonlinear spectral analysis, data compression, image classification, and novel view synthesis from a single 2D rendering of a 3D object.

**Strengths:**

* The paper has a clear motivation
* The proposed method is accompanied with interesting theoretical results
* Limitations are properly discussed
* Codes are provied to support the main paper

**Weaknesses:**

* Overall, the paper is well written but it probably needs more checks for possible typo errors (there is a minor typo at the end of Section 2 "scaler-value")

**Questions:**

1. With respect to the work of Miyato et al., (2022) that considers the properties of the learned model to be intra-orbital homogeneity or full equivariance, which of these properties that the proposed model owns in each setting (U-NFT, G-NFT, g-NFT) ?

---

> ### Author Response · Authors · 2023-11-16
> **Thank you very much for the feedback.**
>
> Thank you very much for the positive feedback and comments. Thank you for pointing out the typography; we checked again the manuscript and fixed the typos in the revision.
>
> In U-NFT and G-NFT setting, the trained model shall share the same property as in MSP(Miyato et al) in terms of intra-orbital homogeneity and inter-orbital homogeneity, given the same condition regarding the continuity and the compactness of the group assumed in their statements.
> When the algorithm “finds' ' one equivariant solution with invertible autoencoder, however, our theory (in particular, Theorem 4.1) does guarantee that it is the unique equivariant solution up-to G-isomorphism.
> Meanwhile,  g-NFT setting has the guarantee of enjoying the full equivariance when the datasets are sufficiently large, because g-NFT pre-defines the representation to be shared across different orbits.

---

> > ### Comment · Reviewer_XpFk · 2023-11-22
> >
> > I thank the authors for their adequate answer. I keep my original rating.

---

### Official Review · Reviewer_uMSh · 2023-11-07

**Soundness:** 3 good
**Presentation:** 2 fair
**Contribution:** 2 fair
**Rating:** 6
**Confidence:** 4

**Summary:**

The main motivation behind the paper is the following: In pre-dominant approaches to equivariant learning, the underlying group and group action is assumed to be known. There are various approaches which then construct linear-equivariant layers followed by a suitable non-linearity and then stack them together. However, often the underlying group or action might not be known. The starting point of the paper is to consider the usual (and then the group) Fourier transform and its well-known equivariance relation. The idea is to construct a non-linear extension where the underlying action is not known, but has similar behaviour as the usual FT. This method dubbed as the non-linear Fourier transform affords finding a relation similar to the usual Fourier transform without having access to analytically tractable knowledge of the group action. It however, does require access to transformed versions of the data to set up a loss function that might be minimized. Various basic properties of this transform and shown. The work heavily relies on ideas from Miyakato et al. and Keller and Welling.

**Strengths:**

- The idea is quite natural and convincing. It does seem more suitable for time series and video data, due to the lack of availability of suitably transformed data in usual setting, but for a framework it seems reasonable.
- It relies on the natural idea of using an invariant kernel mapping (while not cited, there is quite some work on non-neural variants of this, such as by Reisert and Bukhardt, JMLR 2007), and shows how the idea is sensible.
- The idea of finding a data-dependent spectrum is appealing.

**Weaknesses:**

-- The primary weakness of this paper is that it relies on data triplets of tuples that gives a sense of the underlying group action. This is a weakness, since such data isn't easily available in most real-world settings. This is also the reason why the paper pivots to applications in time series in the experimental section.
-- The applications presented don't seem compelling, and more or less follow a similar pattern as in the work of Miyato et al. 2022.
-- Related to the above; The experimental section is relatively toy, which is a weakness, given the generality of the pitch of the paper.

Overall I am just not clear about the applicability of the method.

**Questions:**

Would appreciate if the authors could elaborate on the weaknesses mentioned.

---

> ### Author Response · Authors · 2023-11-17
> **Thank you very much for the feed back and comments!**
>
> Thank you very much for the feedback.
> We would like to make several comments in response to the concern below about the applicability of our method.
> > The primary weakness of this paper is that it relies on data triplets of tuples that gives a sense of the underlying group action. This is a weakness, since such data isn't easily available in most real-world settings. This is also the reason why the paper pivots to applications in time series in the experimental section.
>
> -  Datasets with underlying group action
>
>
> Indeed, the theory of NFT requires that there is some group action that can describe the transformation between a pair of observations, and NFT learns the structure derived from the underlying group.
> We believe that such datasets may occur in nature, for example, when the dataset is derived from 3D space;  much structure of such datasets may be described implicitly by SE(3) or SO(3),  just like how translations-based structures in the natural images play an important role as evidenced by the success of CNN.
>
> Also, as mentioned in the related works section (Sec 6), Koopman-operator-based approaches are much related to our framework as well especially when the transitions are invertible, and applications to the physical systems investigated by these methods are also within the scope of NFT.
>
>
> - Scope of Datasets
>
>
> We would like to note that, depending on the level of inductive bias, we do not necessarily need a dataset that is of time series form. For example, in the example of 2D rendering of 3D objects (Sec 5.2), we are using a pair $(x , g\circ x)$ with the information of $g$.  This type of information is available in Sim2Real situations, for example. We believe that we can also obtain this type of dataset from experiments concerned with control, such as those involving robots.
> We would also like to point out that the task in the rendering example is not necessarily of time series as well, because the task here is to predict a rendering from a given angle.
>
> >While not cited, there is quite some work on non-neural variants of this, such as by Reisert and Bukhardt, JMLR 2007
>
> Thank you for pointing out the related result; however, we did indeed cite this work in the related work section (Section 6)!

---

### Meta-Review · Area_Chair_935P · 2023-12-05

**Metareview:**

A paper proposing an approach (NFT), extending Miyato et al. 2022, for learning latent linear actions of groups on a suitable feature space with varying levels (explicit) knowledge of the group action.

Opinions were somewhat divided as to how well does the experimental section demonstrate the applicability of the proposed method, e.g., considering the reliance on data triplets required in real-world settings. However, all seem to have found the problem tackled in the paper well-motivated, and appreciated the proposed framework, along with the accompanied theoretical results, interesting and appealing. The authors are encouraged to incorporate the important feedback given by the knowledgeable reviewers.

**Justification For Why Not Higher Score:**

Need further evidence for applicability

**Justification For Why Not Lower Score:**

Introduces some new ideas

---

### Decision · Program_Chairs · 2024-01-16

Accept (poster)